# Osteogenic growth peptide is a potent anti-inflammatory and bone preserving hormone via cannabinoid receptor type 2

Bitya Raphael-Mizrahi[1]*, Malka Attar-Namdar[2], Mukesh Chourasia[3], Maria G Cascio[4], Avital Shurki[3], Joseph Tam[3], Moshe Neuman[2], Neta Rimmerman[5], Zvi Vogel[5], Arie Shteyer[6], Roger G Pertwee[4], Andreas Zimmer[7], Natalya M Kogan[2]†, Itai Bab[2]†, Yankel Gabet[1]*†

[1]Department of Anatomy & Anthropology, Sackler Faculty of Medicine, Tel Aviv University, Tel Aviv, Israel; [2]Bone Laboratory, Institute of Dental Sciences, Faculty of Dental Medicine, Hebrew University of Jerusalem, Jerusalem, Israel; [3]Institute for Drug Research, Faculty of Medicine, Hebrew University of Jerusalem, Jerusalem, Israel; [4]Institute of Medical Sciences, University of Aberdeen, Aberdeen, United Kingdom; [5]Department of Neurobiology, Weizmann Institute of Science, Rehovot, Israel; [6]Department of Oral and Maxillofacial Surgery, Hadassah-Hebrew University Hospital, Jerusalem, Israel; [7]Institute of Molecular Psychiatry, University of Bonn, Bonn, Germany

*For correspondence:
bityar@tau.ac.il (BR-M);
yankel@tauex.tau.ac.il (YG)

†These authors contributed equally to this work

**Abstract** The endocannabinoid system consists mainly of 2-arachidonoylglycerol and anandamide, as well as cannabinoid receptor type 1 and type 2 (CB2). Based on previous studies, we hypothesized that a circulating peptide previously identified as osteogenic growth peptide (OGP) maintains a bone-protective CB2 tone. We tested OGP activity in mouse models and cells, and in human osteoblasts. We show that the OGP effects on osteoblast proliferation, osteoclastogenesis, and macrophage inflammation in vitro, as well as rescue of ovariectomy-induced bone loss and prevention of ear edema in vivo are all abrogated by genetic or pharmacological ablation of CB2. We also demonstrate that OGP binds at CB2 and may act as both an agonist and positive allosteric modulator in the presence of other lipophilic agonists. In premenopausal women, OGP circulating levels significantly decline with age. In adult mice, exogenous administration of OGP completely prevented age-related bone loss. Our findings suggest that OGP attenuates age-related bone loss by maintaining a skeletal CB2 tone. Importantly, they also indicate the occurrence of an endogenous peptide that signals via CB2 receptor in health and disease.

## Editor's evaluation

By combining pharmacological and mouse genetic genetics strategies the authors show a clear interaction between the cannabinoid receptor CB2 and osteogenic growth peptide (OGP) in the control of bone remodeling and bone mass. They document that OGP attenuates bone loss by maintaining a skeletal CB2 tone, and it does so by allosterically binding to the CB2 receptor. These novel observations should allow further investigations on cannabinoid based strategies for skeletal diseases.

## Introduction

In the past two decades, the endocannabinoid (EC) system was discovered and characterized, after identifying receptors for $\Delta^9$-tetrahydrocannabinol (THC), the major psychoactive component of marijuana and hashish. The actions of THC are mediated mainly by the cannabinoid receptor type

1 (CB1) and type 2 (CB2) (*Mackie, 2006*). Both are seven-transmembrane domain, class A G-protein-coupled receptors (GPCRs) sharing 44% overall identity (68% similarity considering the transmembrane regions alone) (*Munro et al., 1993*). CB1 is expressed predominantly in neurons and to a lesser extent in several other tissues. In health, CB2 is expressed mainly in immune and bone cells (*Mackie, 2005*). Owing to the relatively high expression of CB2 in immune cells, it has been hypothesized that this receptor mediates the immunosuppressive effects of phyto- and synthetic cannabinoids (*Munro et al., 1993*). It is widely believed that CB1 and CB2 are principally targeted by two lipid-derived endogenous ligands, N-arachidonoylethanolamine (AEA or anandamide) and 2-arachidonoylglycerol (2-AG) (*Bab et al., 2008*). These ECs derive from arachidonic acid and are present mainly in the brain as well as in a variety of peripheral tissues (*Bab et al., 2009*). Quantitative wise, the main EC is 2-AG.

It has been suggested that 2-AG and AEA are synthesized and function 'on demand' (*Hashimotodani et al., 2013*) because (i) non-stimulated CB2-deficient mice have a normal phenotype (*Buckley et al., 2000*) and (ii) the ECs are rapidly degraded (*Blankman and Cravatt, 2013*). By contrast, studies in mice and humans have established that the functional activity of CB2 protects the skeleton against age-related bone loss (*Ofek et al., 2006*; *Idris et al., 2008*; *Karsak et al., 2005*), suggesting the existence of a 'CB2 tone' maintained by more stable ligands. Other findings provide room for such an additional agonist. Pharmacological studies dealing with anandamide and 2-AG binding at CB2 reported rather low binding affinities at the μM range, more specifically, one order of magnitude lower than that of THC (*Munro et al., 1993*; *Mechoulam et al., 1995*), suggesting that there could be a more specific and potent endogenous ligand. Furthermore, physiological results from several research groups suggest that anandamide and 2-AG have opposing effects on CB2 signaling (*Tam et al., 2008*; *Gonsiorek et al., 2000*). In this study, we aimed at finding a putative additional endogenous specific and potent CB2 peptidic agonist that maintains a CB2 tone throughout life and protects against bone loss.

Many proteins and peptides were shown to be GPCR ligands that play a major role in modulating GPCR activation and expression via direct crosstalk. Targeting of GPCRs and other receptors by non-peptidic lipophilic ligands (e.g. acetylcholine) results in short-term activation, whereas the tonal activity of some of these receptors (e.g. muscarinic and nicotinic) is maintained by protein or peptide ligands with higher binding affinities (*Alreja et al., 2000*; *Wu and Yeh, 2005*; *Miwa et al., 2012*). In GPCRs, small-molecule lipophilic ligands have their binding pocket in transmembrane loops, whereas proteins and peptides bind to the extracellular domains (*Hurst et al., 2013*). Indeed, previous studies have shown the existence of a series of endogenous peptide cannabinoid ligands, termed 'pepcans' (*Bauer et al., 2012*). One of them, 'pepcan-12,' is a 12-amino-acid peptide exhibiting a binding affinity to CB1 in the nanomolar range. Consistent with the binding of peptides and lipophilic ligands to different domains, pepcan-12 potently targets an allosteric site at CB1 and interacts rather weakly with the 'classical' CB1 binding pocket, characterized by the high-affinity binding of the synthetic cannabinoids CP55940 and WIN55212–2 (*Bauer et al., 2012*). This same peptide was later identified as a CB2 positive allosteric modulator (PAM) (*Petrucci et al., 2017*). Pepcan-12 has been identified at significant levels in the brain, peripheral tissues, and plasma following ischemia/reperfusion injury, suggesting its pathophysiological role in modulating CB2 activation (*Petrucci et al., 2017*). However, pepcan-12 has no reported actions in bone cells and does not seem to decline with age, thus bringing into question its involvement in preserving bone mass during aging.

To assess the possibility that the skeletal protective actions of CB2 are maintained by an endogenous peptide agonist, we performed an in-depth literature search, looking for a peptide/protein that can bind to CB2 or to any other GPCR in bone cells. In January 2017, a search in PubMed looking for '(peptide or protein) and ('Gi protein' or 'G(i)' or 'GPCR' or 'G*coupled receptor') and (osteoblast* or osteoclast* or osteocyt*)' resulted in 96 articles (*Figure 1A*). Out of these 96 articles, there were only 26 Gi agonists and 10 were endogenous proteins/peptides (*Supplementary file 1*). Furthermore, 4 out of the 10 endogenous peptides/proteins were reported to bind to an unknown Gi or to CB2 in bone cells (*Supplementary file 1*). Out of these four remaining peptides/proteins, two were affected by aging and only one, named osteogenic growth peptide (OGP), played a positive role in maintaining bone mass (*Spreafico et al., 2006*; *Bab et al., 1992*; *Greenberg et al., 1993*; *Greenberg et al., 1995*). We could not find any report on the putative receptor for OGP. We therefore proceeded with this peptide to assess whether it binds to and activates CB2.

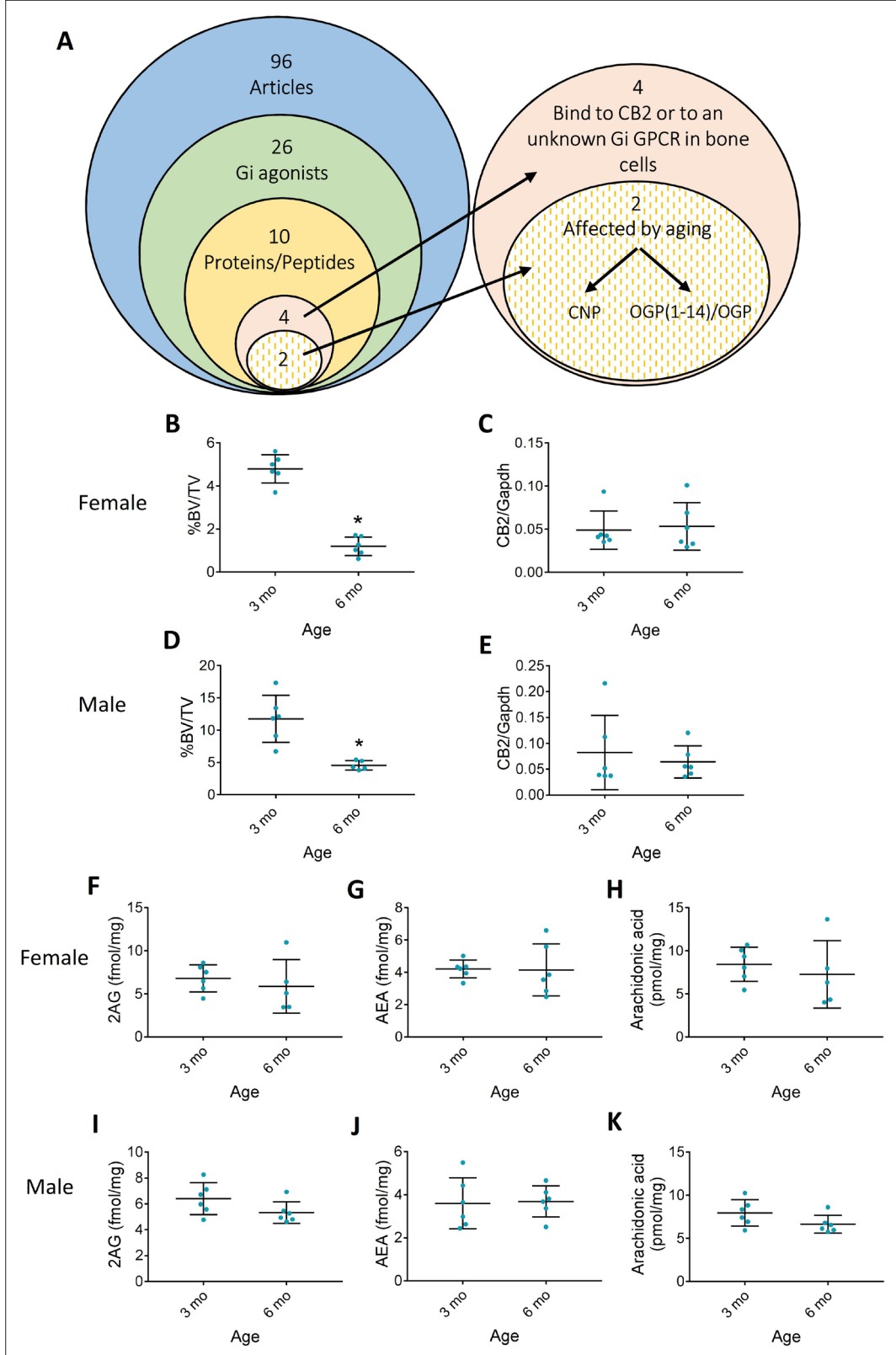

**Figure 1.** Aging differences in the endocannabinoid (EC) system. (**A**) Literature scan isolated osteogenic growth peptide (OGP) as the only peptide that is mitogenic in osteoblasts and whose levels decline with age. (**B–E**) Age-related bone loss (decrease in trabecular bone volume density, %BV/TV) is not associated with changes in cannabinoid receptor type 2 (CB2) expression levels in 6 month (6 mo) vs. 3-month-old (3 mo) female (B and

*Figure 1 continued on next page*

*Figure 1 continued*

C, respectively) and male (D and E, respectively) mice. (**F–K**) Levels of EC ligands in the bone tissue of aging 6 mo vs. 3 mo female (F, G, and H) and male mice (I, J, and K). Data are the mean ± SD; n=6; *p<0.05 vs. 3 mo mice, t-test.

The online version of this article includes the following source data and figure supplement(s) for figure 1:

**Source data 1.** BV/TV female and male mice 6 and 3 month old.

**Source data 2.** Cnr2 expression male and female 3 and 6 month mice.

**Source data 3.** Endocannabinoid levels in 3 and 6 month old mice.

**Figure supplement 1.** Age-related differences in femoral bone.

---

In 1992, we reported that a C-terminal 14-amino acid peptide derived from the gene for histone H4, H4(90–103) in mice and H4(89–102) in humans, is mitogenic to osteoblasts (*Spreafico et al., 2006*; *Bab et al., 1992*; *Greenberg et al., 1993*), stimulates bone formation in rodents and rescues ovariectomy-induced bone loss (*Bab et al., 1992*; *Chen et al., 2000*). This peptide was therefore called OGP(1-14). OGP(1-14) is physiologically present in the serum at nano- to micromolar concentrations, with most of it complexed to α2-macroglobulin (*Greenberg et al., 1995*; *Gavish et al., 1997*; *Bab et al., 1999*). Upon dissociation from this complex, it is proteolytically cleaved, thus generating its 5-amino-acid cellular activator OGP (*Idris et al., 2008*; *Karsak et al., 2005*; *Mechoulam et al., 1995*; *Tam et al., 2008*; *Gonsiorek et al., 2000*) referred to here as OGP (*Bab et al., 1999*). This pentapeptide targets a pertussis toxin-sensitive GPCR and consequently activates a Gi-protein – Erk1/2 – Mapkapk2 – CREB cascade (*Miguel et al., 2005*). Although OGP has never been attributed anti-inflammatory properties (a hallmark of CB2 agonists), there is a striking resemblance between the signal transduction triggered by this pentapeptide and that of CB2 agonists in bone cells (*Ofek et al., 2011*). Moreover, CB2 agonists and OGP have an identical mitogenic dose-response activity in osteoblasts and can stimulate bone formation and bone mass in intact rats and osteoporotic mice (*Idris et al., 2008*; *Bab et al., 1992*; *Chen et al., 2000*; *Ofek et al., 2011*). Interestingly, the serum levels of biologically active OGP decline with age during adulthood (*Greenberg et al., 1995*). We therefore hypothesized that OGP is a CB2 agonist, and that its serum levels drive an age-related reduction in CB2 tone, thus alleviating the protective role of CB2 during aging. Indeed, in a series of ex vivo and in vivo skeletal and inflammatory experiments, we show here that OGP shares the activities of highly potent CB2 agonists and that these actions depend on the presence of an active CB2 receptor. Our present results suggest for the first time that OGP is an endogenous peptide agonist at CB2.

## Results

### Age-related changes in CB2 and its classical agonists

Bone loss is an inherent consequence of aging in both mice and humans (*Bab et al., 2007*; *Bonnick, 2006*). Because CB2 expression was shown to protect against age-related bone loss (*Ofek et al., 2006*), we first investigated whether age induces a decline in CB2 expression or the levels of its classical endogenous agonists in wild type (WT) animals. We therefore examined the skeletal changes in male and female mice between the ages of 3 and 6 months, when a dramatic age-related bone loss occurs (*Bab et al., 2007*), and measured the bone levels of AEA, 2AG, and arachidonic acid as well as CB2 expression in 3 vs. 6-month-old male and female mice. As expected, our results show a significant age-related decline in bone mass in both sexes (*Figure 1B and D* and *Figure 1—figure supplement 1*). In 3 months, the trabecular BV/TV decreased by 61 and 75% in males and females, respectively, due to significant 64 and 70% decrease in the bone volume. Importantly, we did not find any difference in CB2 expression (*Figure 1C and E*) or in AEA, 2AG, and arachidonic acid tissue levels in either gender (*Figure 1F–K*). These results support our working hypothesis that another endogenous agonist is responsible for maintaining long-term CB2 activation, and that age-related bone loss is caused by a decline in the levels of this agonist.

### OGP binds CB2 and activates intracellular signaling

Attenuation of forskolin (FSK)-stimulated cAMP levels is a hallmark of Gi-coupled CB2 activation (*Slipetz et al., 1995*). Like cannabinoid receptor agonists, OGP displays this activity in CHO cells

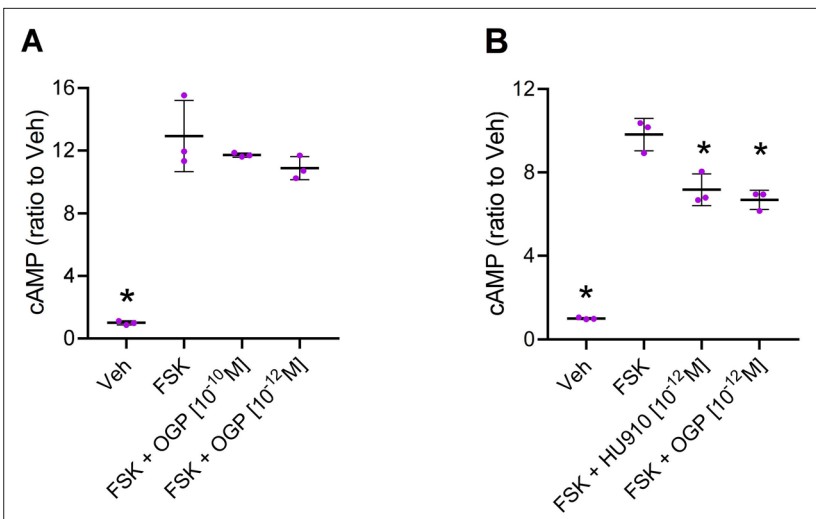

**Figure 2.** Osteogenic growth peptide (OGP) attenuates forskolin (FSK)-stimulated cAMP levels via cannabinoid receptor type 2 (CB2). (**A**) CHO cells mock transfected with an empty vector (that does not express CB2) were treated with vehicle (Veh) or forskolin (FSK) to induce cAMP levels (with Veh only). FSK-treated cells were also pre-treated with two concentrations of OGP. (**B**) CHO cells transfected with human *CNR2* received Veh only (Veh) or were treated with FSK and pre-treated with Veh (FSK), the CB2 agonist HU910 (FSK +HU910) or OGP (FSK +OGP). Data are the mean ± SD obtained in triplicate culture wells per condition and repeated three times. *p<0.05 vs. FSK alone, non-parametric one-way ANOVA.

The online version of this article includes the following source data and figure supplement(s) for figure 2:

**Source data 1.** Mock and hCB2-CHO cAMP levels (binding assay).

**Source data 2.** cAMP levels in hCB2-CHO with CB2 antagonist (SR2).

**Source data 3.** hCB1-HEK293 cAMP levels.

**Figure supplement 1.** Osteogenic growth peptide (OGP) inhibits the forskolin-stimulated cAMP in CHO-hCB2 cells.

**Figure supplement 2.** Osteogenic growth peptide (OGP) does not activate cannabinoid receptor type 1 (CB1) internal signaling in hCB1-HEK293 cells.

expressing human CB2 (*Figure 2* and *Figure 2—figure supplement 1*), demonstrating that OGP binds to and activates CB2. However, OGP was unable to affect cAMP levels in HEK293 cells expressing human CB1 (*Figure 2—figure supplement 2*), supporting the notion that OGP preferentially binds to CB2 rather than CB1 at the studied working concentrations. Moreover, we found that inhibition of FSK-stimulated activity in CHO-hCB2 by OGP displayed a bell-shaped dose-response curve with half-maximal inhibition at logEC50=−12.10 in two independent experiments. The addition of the CB2 antagonist SR144528 (1 μM) to OGP resulted in a right-shifted bell-shaped curve with a half-maximal inhibition at logEC50=−10.98 and −9.93 in two independent experiments (*Figure 2—figure supplement 1*).

## CB2-OGP docking followed by molecular dynamics (MD) simulation

Docking studies of OGP to both the active and inactive models of CB2 were used to predict feasible binding sites of OGP in CB2. These studies did not rule out binding of OGP at the orthosteric domain but suggested another potential binging site - possibly an allosteric site - at the extracellular region of CB2. The latter, while not being identical in the two receptor models, was very similar and involved in both cases mainly the N-terminal, ECL1(Extra-Cellular Loop), and ECL2 (*Supplementary file 2*). Furthermore, as one would expect from a receptor agonist, docking calculations suggest that OGP exhibits higher affinity to the active model compared with the inactive model (*Supplementary file 2*). These studies also suggested that OGP and CP55940 bind to CB2 simultaneously, at distinct sites without reciprocally interfering. Indeed, MD simulations with OGP at the proposed binding site within the ECL region show that the peptide stably stays in that region (*Figure 3A*) with no noticeable

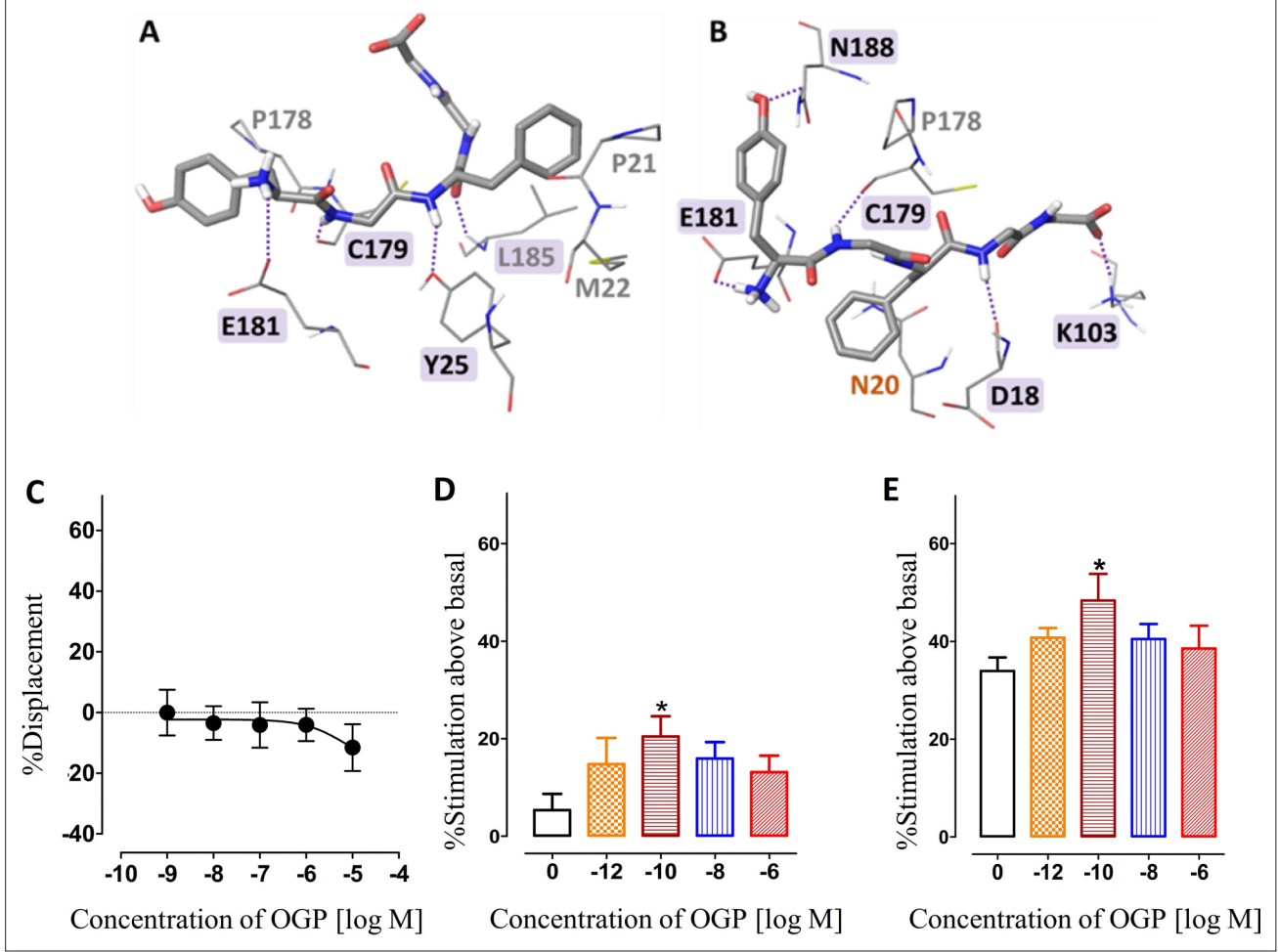

**Figure 3.** Interaction of the osteogenic growth peptide (OGP) with the allosteric site of human cannabinoid receptor type 2 (CB2). (**A**) and (**B**) represent conformation of the peptide when OGP is bound to the extracellular surface in the absence (A) and presence (B) of CP55940. The conformations presented are taken from the last step of the molecular dynamics (MD) simulation. Residues forming direct hydrogen bonds with the peptide and the respective interactions are highlighted in purple. Residues forming hydrophobic interactions are labeled in gray color. The amino-π interaction is labeled in orange color. (**C**) Competitive binding assay between OGP and tritiated CP55940. CHO-CB2-derived membranes were used for a CB2 binding assay. No detection of CP55940 chase-out in the presence of increasing concentrations of OGP indicates that OGP and CP55940 do not compete over the same binding site at CB2. (**D,E**) Effect of OGP on the stimulation of binding of [35S]GTPγS induced by CP55940. CHO-CB2-derived membranes were used in a [35S]GTPγS binding assay. The white bar shows the relative activation of CB2 by 0.1 nM (D) or 10 nM (E) of CP55940 above basal levels (DMSO(Dimethyl sulfoxide) only) in the absence of OGP. Other graphs show relative activation of CB2 induced by CP55940 in the presence of increasing concentrations of OGP. *, $p < 0.05$, vs. CP55940 only (no OGP).

The online version of this article includes the following figure supplement(s) for figure 3:

**Figure supplement 1.** Molecular dynamics (MD) energy image.

changes in the dynamic energy even in the presence of the classical agonist CP55940 (*Figure 3B* and *Figure 3—figure supplement 1*).

## OGP acts as a PAM in the presence of a lipophilic CB2 agonist

Following up on our docking simulation that supported the notion of a stable binding of OGP in the presence of a lipophilic agonist in the classical intramembranous binding site, we performed here a competition assay between tritiated CP55940 a synthetic lipophilic CB2 agonist, and OGP. Our results showed that OGP did not displace CP55940, suggesting that there is no competition between these agonists at the same binding site (*Figure 3C*). In the absence of competition at the orthosteric binding site, we next asked whether OGP binding at an allosteric site modulates the functional activity of orthosteric agonists. To this end, we treated hCB2-CHO-derived membranes with CP55940 and

measured receptor activation using the GTPγS binding assay (*Figure 3D and E*). Notably, increasing concentrations of OGP up to $10^{-10}$ M resulted in significantly higher CB2 activation by CP55940 as indicated by the stimulation of $^{[35S]}$GTPγS induced by 0.1 nM CP55940 (*Figure 3D*) and 10 nM CP55940 (*Figure 3E*). These results further support the notion that OGP binds at an allosteric site at CB2 and indicate that OGP acts as a PAM in the presence of lipophilic CB2 agonists.

## CB2 is essential for maintaining the proliferative activity of OGP in murine and human osteoblasts

It has been repeatedly demonstrated that the effect of OGP on osteoblast proliferation exhibits a biphasic relationship, characterized by a dose-response stimulation at low peptide concentrations, followed by reversal of this enhancement at higher doses (*Miguel et al., 2005*; *Bab and Chorev, 2002*). A similar biphasic effect was also reported for cannabinoid ligands (*Ofek et al., 2006*; *Ofek et al., 2011*; *Smoum et al., 2015*) and for other ligands of other GPCRs (*Schattauer et al., 2019*). Indeed, we show here in mouse-derived cultures that the osteoblastic proliferative activity of both OGP and HU-910, a potent CB2-selective agonist (*Horváth et al., 2012*), is characterized by nearly an identical dose response relationship (*Figure 4A and B*). More importantly, the activity of OGP is completely abrogated following genetic ablation of CB2, as demonstrated using cell count and the bromodeoxyuridine (BrdU) assay (*Figure 4B* and *Figure 4—figure supplement 1*, respectively). The proliferative effect of OGP in both murine (*Figure 4—figure supplement 2*) and human osteoblasts (*Figure 4C and D*) was also abrogated following pharmacological blockade with the CB2-selective antagonist SR144528. These results indicate that the presence of CB2 is essential for OGP activity in osteoblasts.

## CB2 mediates the OGP attenuation of osteoclast differentiation

CB2 agonists inhibit osteoclast differentiation in ex vivo cultures of bone marrow monocytes in the presence of M-CSF and RANKL (*Ofek et al., 2006*). We show here that OGP shares the same anti-osteoclastogenic activity in cultures derived from WT animals (*Figure 4E and F*). By contrast, this effect of OGP is entirely absent in *Cnr2*$^{-/-}$-derived cultures (*Figure 4E and F*), demonstrating that the expression of CB2 is essential for mediating the anti-osteoclastogenic effect of OGP.

## CB2 mediates the bone-protective effect of OGP in vivo

The anabolic effect of OGP and other CB2 agonists in ovariectomized (OVX) mice was reported in separate studies (*Ofek et al., 2006*; *Chen et al., 2000*; *Smoum et al., 2015*). Here, we employed the OVX-induced bone loss model where mice were treated with OGP 6 weeks post-OVX. As expected, the vehicle (Veh) treated OVX animals (both WT and *Cnr2*$^{-/-}$) showed a significant bone loss in the trabecular bone compartment compared with the Sham-OVX control animals (*Figure 5*). There were no significant OVX-induced changes in the cortical bone of the WT mice, but in the *Cnr2*$^{-/-}$ mice, OVX induced slight but statistically significant decrease in volumetric bone mineral density, diaphyseal diameter, and cortical area over total bone area (–5, –3, and –5%, respectively, vs. Sham, *Figure 5K and L* and *Figure 5—figure supplement 1*). Daily treatment of OVX mice with OGP significantly restored BV/TV in the trabecular bone compartment (+45.9% vs. OVX, Veh-treated mice, p=0.035, *Figure 5A*) of WT animals as previously described (*Bab and Chorev, 2002*). This was accompanied with significant increases in trabecular connectivity density and trabecular number (Tb.N) but not trabecular thickness (Tb.Th) (*Figure 5B, C and D*). OGP did not affect any of the cortical bone parameters in the WT mice (*Figure 5E–F and K–L* and *Figure 5—figure supplement 1* A-D). Importantly, OGP had no effect on any of the trabecular and cortical bone parameters in OVX *Cnr2*$^{-/-}$ animals (*Figure 5G–L*). This observation demonstrates that the bone protective capacity of OGP is mediated by CB2.

## CB2 mediates the anti-inflammatory activity of OGP in macrophages

Anti-inflammatory activity is a hallmark of CB2 agonists (*Sun et al., 2017*; *Ossola et al., 2016*; *Malfitano et al., 2014*). Bone marrow-derived macrophages (BMDMs) obtained from WT and *Cnr2*$^{-/-}$ mice were cultured for 24 hr with M-CSF. Bacterial lipopolysaccharide (LPS) was added to stimulate inflammation (*Ngoh et al., 2016*) in all the cultures, excluding the negative control. OGP was added at increasing concentrations before adding LPS to the cultures. Gene expression of pro-inflammatory

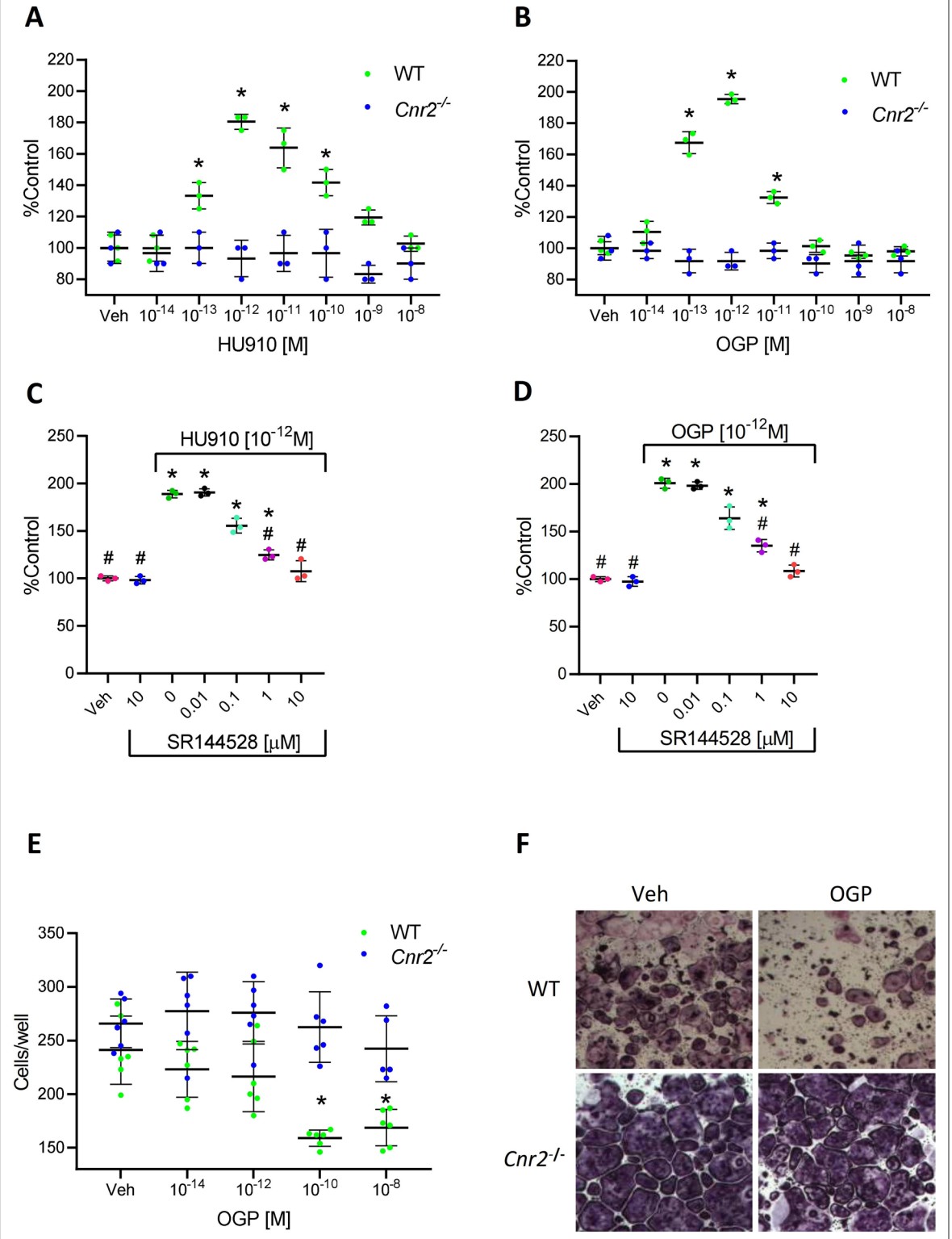

**Figure 4.** The effect of osteogenic growth peptide (OGP) in bone cells is dependent on cannabinoid receptor type 2 (CB2). (**A–B**) Proliferative activity of HU910, a synthetic CB2 selective agonist (A), and of OGP (B) in wild type (WT) and *Cnr2⁻ᐟ⁻*-derived murine osteoblasts. Data are the mean ± SD obtained in triplicate and were repeated at least two times. *p<0.05 vs. vehicle (Veh) in the same genotype, non-parametric oneway ANOVA. (**C–D**) The proliferative activity of HU910 (C) and OGP (D) in human osteoblasts is abrogated by the CB2-selective antagonist SR144528 (SR2). Data are mean ± SD obtained in triplicate. *p<0.05 vs. Veh, #p<0.05 vs. the CB2 agonist with no SR2, non-parametric oneway ANOVA. (**E–F**) OGP attenuates

*Figure 4 continued on next page*

*Figure 4 continued*

osteoclastogenesis (number of multinucleated tartrate-resistant acid phosphatase [TRAP] positive cells per well) in WT- but not in *Cnr2*[-/-]-derived murine cultures. Data are the mean ± SD obtained in six wells per condition. *p<0.05 vs. Veh in the same genotype, non-parametric one-way ANOVA.

The online version of this article includes the following source data and figure supplement(s) for figure 4:

**Source data 1.** Murine WT and Cnr2-/- osteoblasts proliferation.

**Source data 2.** Human osteoblasts proliferation with CB2 antagonist.

**Source data 3.** Murine WT and Cnr2-/- osteoclasts differentiation.

**Source data 4.** BrdU on WT osteoblasts with OGP.

**Source data 5.** BrdU on Cnr2-/- osteoblasts with OGP.

**Source data 6.** Murine osteoblasts with OGP, HU910 and CB2 antagonist.

**Figure supplement 1.** The effect of osteogenic growth peptide (OGP) in murine bone cells is dependent on CB2.

**Figure supplement 2.** The effect of osteogenic growth peptide (OGP) in murine bone cells is dependent on cannabinoid receptor type 2 (CB2).

cytokines (*Il1b* and *Tnfa*) indicated that LPS induces a dramatic inflammatory response, and that OGP completely abrogates the expression of these cytokines at $10^{-14}$ to $10^{-12}$ M doses (*Figure 6A and B*). Notably, the effect of OGP was absent in *Cnr2*[-/-] cells (*Figure 6C and D*), indicating that the anti-inflammatory effect of the peptide is mediated by CB2.

## CB2 mediates the anti-inflammatory activity of OGP in vivo

We then investigated whether OGP produces the same anti-inflammatory effects as CB2 agonists in vivo. We used the mouse ear-swelling model, which is characterized by a rapid and significant acute inflammatory response to xylene irritation (*Smoum et al., 2015*). In this model, ear swelling is completely prevented by administering the COX inhibitor indomethacin (*Figure 7*). Consistent with the established inhibition of mast cell degranulation by CB2 agonists (*Jonsson et al., 2006*), OGP significantly inhibited the effect of xylene on ear swelling (–70%), similar to the CB2-selective agonist HU910 (*Figure 7A and C*). However, in *Cnr2*[-/-] mice, OGP and HU910 had no effect on ear swelling (*Figure 7B and C*). Because indomethacin had the same inhibitory effect in WT and *Cnr2*[-/-] mice, these findings indicate that the effect of OGP is entirely CB2 dependent and distinct from that of prostaglandins.

## Age-related changes in serum levels of OGP and its precursor OGP(1-14) in humans

OGP(1-14), the precursor of OGP, is secreted by stromal cells to the serum, where it binds to α2-macroglobulin (*Gavish et al., 1997*). Upon its release, the 14-amino acid peptide is cleaved into the active pentapeptide. Both are detected by the same antibodies, hence the name immunoreactive (ir) OGP(1-14), which also includes the levels of free OGP. In a previous study, we reported that the ratio between free OGP and total irOGP(1-14) in the serum decreased between the third and the fifth decades of life in men (*Greenberg et al., 1995*). Importantly, this age range is characterized by age-related bone loss in humans, independently of sex hormones (*Demontiero et al., 2012*). Here, we investigated whether irOGP(1-14) serum levels decline with age during the same period in premenopausal women. Our analysis revealed that during the fifth decade of life irOGP(1-14) serum levels are significantly lower than during the third and fourth decades (p=0.039 and p=0.016, respectively, *Figure 8A*). This finding supports the notion that serum OGP(1-14) and OGP levels decline with age, thus contributing to the reduced CB2 tone and related bone loss. Owing to the limited serum volume in mice, we were not able to accurately measure the serum levels of OGP(1-14) or OGP in mice. Nevertheless, to assess the role of OGP in preventing age-related bone loss, we experimentally investigated the skeletal effect of exogenous administration of OGP during adulthood. To this end, we treated 3-month-old male mice with OGP or Veh for 3 months. This age range is characterized by slow age-related bone loss that follows the peak bone mass at 3 months of age (*Figure 1*). Indeed, Veh-treated mice (6 mo-Veh) displayed a significant trabecular bone loss (30% reduced BV/TV, 54% reduced Conn.D, and 36% reduced Tb.N; no difference in Tb.Th) relative to the 3-month-old mice (p=0.01, *Figure 8B–E and H*). The cortical bone did not show significant changes between the age of 3 and 6 months (*Figure 8F–G*, and *Figure 8—figure supplement 1*). Daily administration of 1 µg/kg/

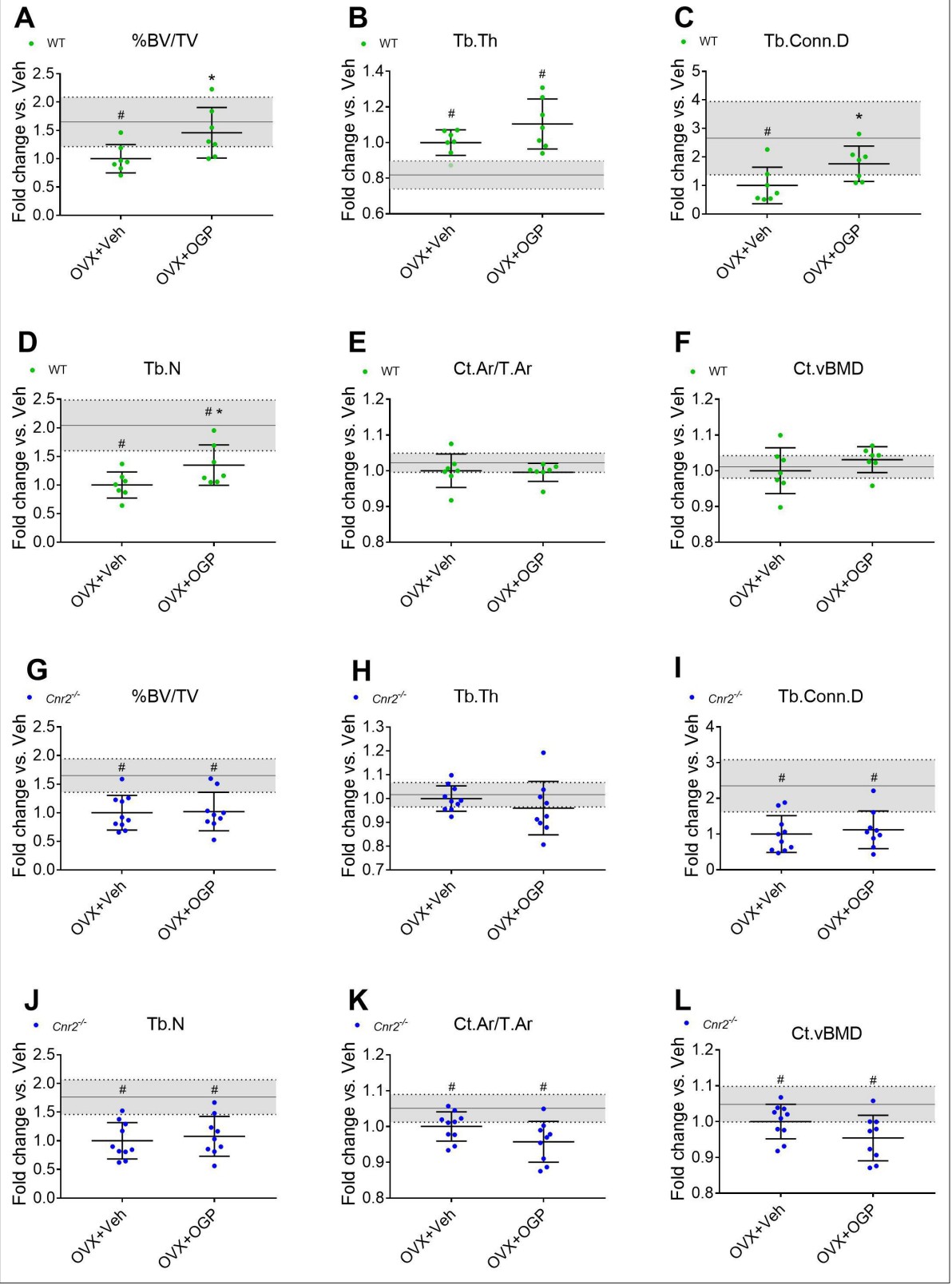

**Figure 5.** Osteogenic growth peptide (OGP) effect on bone recovery in an ovariectomized (OVX) model is cannabinoid receptor type 2 dependent. Rescue of wild type (WT) (**A–F**) but not *Cnr2*-/- (**G–L**) OVX-induced bone loss by OGP administration. 8-week old mice were OVX (or Sham-OVX), and treatment with OGP or vehicle started after 6 weeks for 6 weeks. (A, G) trabecular bone volume fraction (BV/TV); (B, H) trabecular thickness (Tb.Th); (C, I) trabecular connectivity density (Tb.Conn.D); (D, J) trabecular number (Tb.N); (E, K) cortical area over total bone area (Ct.Ar/T.Ar); (F, L) cortical

*Figure 5 continued on next page*

*Figure 5 continued*

volumetric bone mineral density (CT.vBMD). Results obtained from WT Sham-OVX, WT OVX+Veh and WT OVX+OGP (n=7); *Cnr2*⁻/⁻ Sham-OVX, *Cnr2*⁻/⁻ OVX+Veh and *Cnr2*⁻/⁻ OVX+OGP (n=9–10). Data are mean ± SD normalized to the OVX+Veh group. Sham-OVX mean and SD are represented by continuous line and shaded area, respectively. *$p<0.05$ vs. OVX+Veh in the same genotype. #$p<0.05$ vs. Sham-OVX.

The online version of this article includes the following source data and figure supplement(s) for figure 5:

**Source data 1.** MicroCT morphometric data of OVX and Sham-OVX mice.

**Figure supplement 1.** Osteogenic growth peptide (OGP) effect on bone recovery in ovariectomized (OVX) mice is cannabinoid receptor type 2 dependent.

day OGP for 3 months completely prevented the trabecular bone loss (*Figure 8B–E and H*) and had a significant effect on the cortical bone (cortical expansion, *Figure 8—figure supplement 1*). Overall, the OGP-treated mice at 6 months of age displayed bone structural values similar to their 3-month-old counterparts, thus completely preventing the age-related skeletal changes (*Figure 8B–H*).

## Discussion

The CB2 tone has been emerging as a key theme in the homeostatic maintenance of the skeletal and possibly other physiological systems. Insights derived from our studies suggest that the actions of OGP on CB2-regulated processes are at least equally important to those of the arachidonic acid-derived ECs: 2-AG and AEA. Moreover, our data suggest that OGP rather than the classical ECs is essential for maintaining the CB2 tone in bone during aging.

Several studies in the last two decades demonstrated a major role for CB2 in suppressing inflammatory responses and related conditions such as neuropathic pain, ischemia/reperfusion injury, and atherosclerosis, as well as protection against age-related bone loss (*Ofek et al., 2006*; *Racz et al., 2008*; *Prime et al., 2009*; *Steffens et al., 2005*). This role has been revealed primarily by using phyto- or synthetic cannabinoids rather than ECs, which are unstable, short-lived neurotransmitter-like compounds. The activity of such agonists is inconsistent with the continuous tone required for attenuating age-related bone loss. Rather, we hypothesized that circulating peptide agonists at appropriate concentrations fulfill these functions.

We demonstrated here that the bone loss that started during adulthood in mice is neither accompanied by age-related changes in the levels of the two classical ECs nor in CB2 expression levels in bone tissue (*Bab et al., 2007*, *Figure 1*). These results suggest that another physiological CB2 agonist declines with time, thus explaining this age-related bone loss. Our literature survey, performed in 2017, identified OGP(1-14)/OGP as a likely candidate (*Figure 1A*) that is present in the circulation at the micromolar level under physiological conditions (*Bab et al., 1992*; *Greenberg et al., 1995*; *Bab and Chorev, 2002*). The receptor for this peptide has remained unknown for almost three decades, although we showed in separate studies that OGP and CB2 receptor cannabinoid ligands trigger the same intracellular signaling pathway (*Miguel et al., 2005*; *Ofek et al., 2011*). Here, we demonstrate for the first time that OGP binds to and activates CB2 using several in vitro and in vivo assays in the skeletal and immune systems. Interestingly, we previously showed a relative age-related decline of free OGP in human male subjects (*Greenberg et al., 1995*), and here, we show in premenopausal women a significant age-related reduction in the total serum OGP(1-14) and OGP levels (*Figure 8A*).

Virtually all living cells may be able to secrete certain amounts of endogenous AEA or 2-AG, and the serum used in in vitro assays may contain small amounts of ECs. Our quick assays performed in CHO cells with no serum (*Figure 2B*) may suggest that OGP can induce CB2 signaling by itself. Because of its hydrophilic nature and dramatically different molecular structure, OGP is less likely to compete with lipophilic cannabinoid ligands (e.g. AEA and 2-AG) at the classical transmembranous binding site under normal physiological conditions. This lack of competition would suggest that the allosteric binding of OGP at CB2 can modulate the orthosteric binding of lipophilic agonists. Indeed, our docking simulation together with our competitive binding assays between OGP and tritiated CP55940 demonstrate that OGP binds at an allosteric site and modulates the level of CB2 activation by the orthosteric agonist (*Figure 3*). From these results, it is still unclear whether OGP increases CP55940 binding at CB2 or merely the CB2 functional response to CP55940. Be that as it may, these results further support the notion that OGP acts as a PAM.

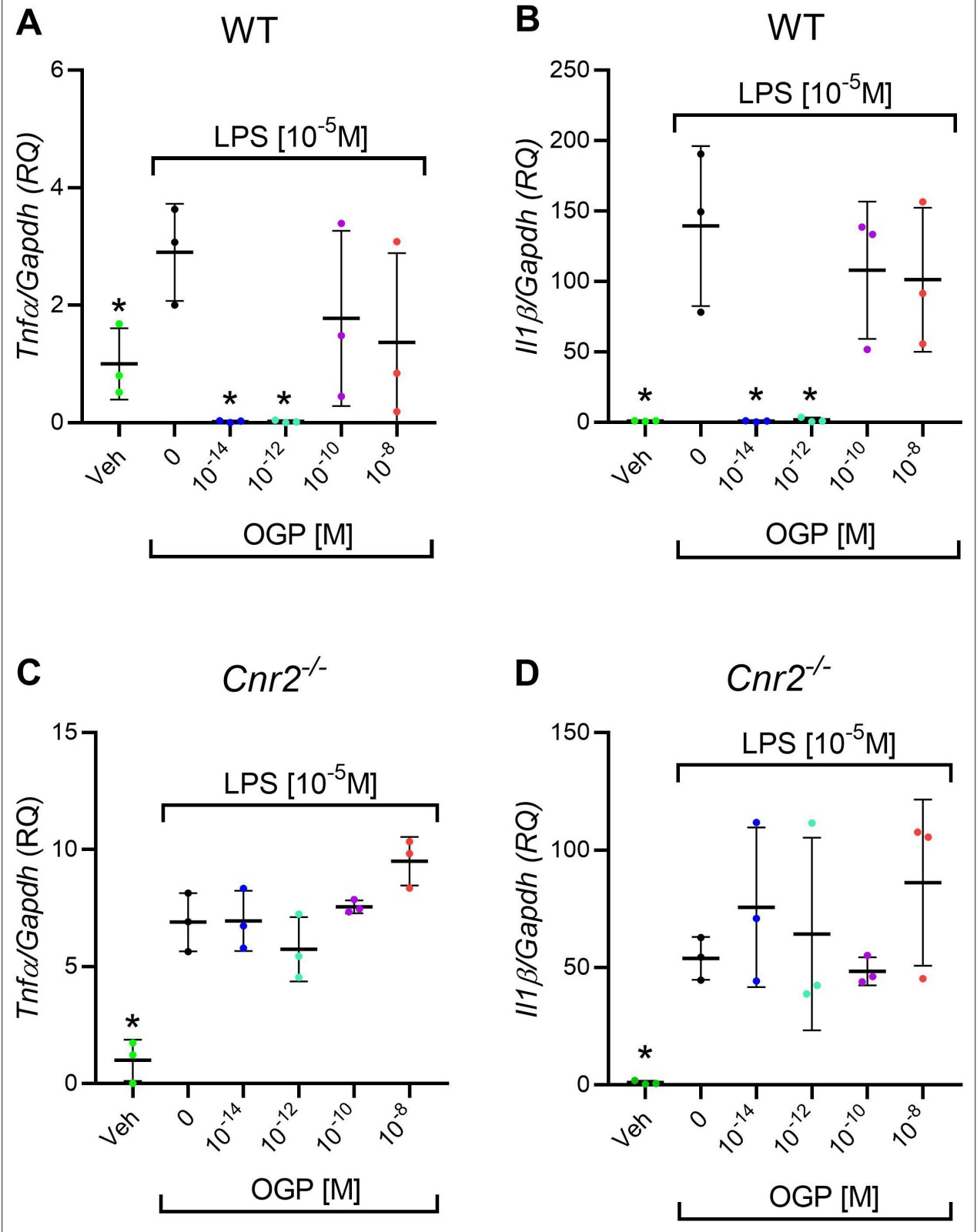

**Figure 6.** In vitro anti-inflammatory activity of osteogenic growth peptide (OGP) in wild type (WT) but not in *Cnr2*[-/-] macrophage cultures. (**A–B**) Lipopolysaccharide (LPS)-induced expression of *Tnfa* (A) and *Il1b* (B) in WT macrophages pre-treated with OGP. (**C–D**) LPS-induced expression of *Tnfa* (**C**) and *Il1b* (**D**) in *Cnr2*[-/-] macrophages pre-treated with OGP. Data are normalized to the vehicle (Veh) value (RQ) and are presented as the mean ± SD obtained in triplicate. *p<0.05 vs. LPS alone, non-parametric one-way ANOVA.

*Figure 6 continued on next page*

*Figure 6 continued*

The online version of this article includes the following source data for figure 6:

**Source data 1.** Murine WT and Cnr2 macrophages cytokine levels.

Computational docking and MD studies suggest a possible allosteric site for OGP binding at the CB2 receptor. While not ruling out the possibility of binding at the orthosteric site, our simulations imply that OGP can also bind to the ECL region and that this complex can stably last for a prolonged time. Docking to both receptor models generated resembling binding and involved similar interactions with residues from the ECL1, ECL2, and the N-terminal. The exact binding mode remains uncertain as the exact structure of the N-terminal, which is crucial to the exact binding mode of an

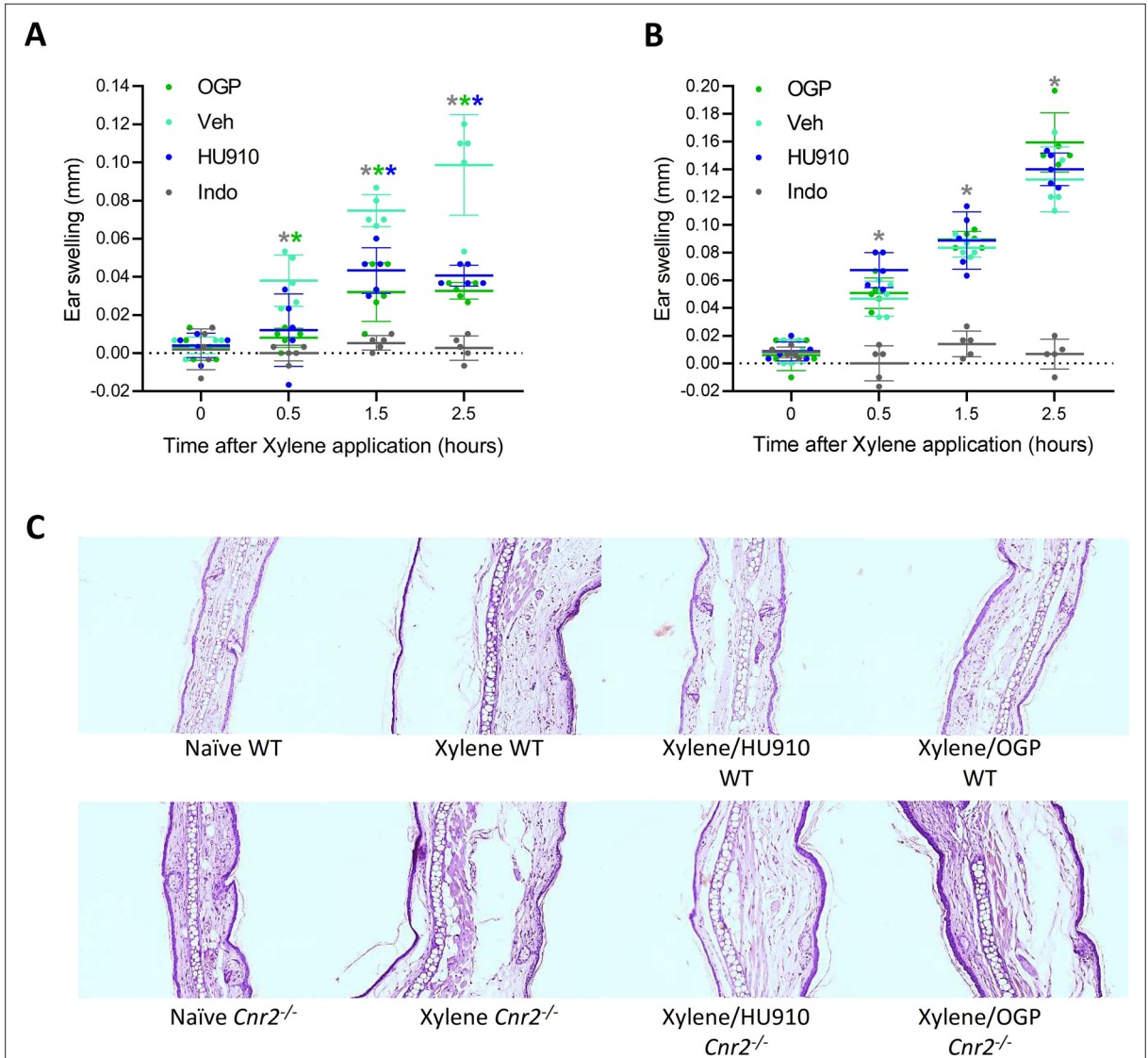

**Figure 7.** Osteogenic growth peptide (OGP) attenuates xylene-induced ear swelling. (**A**) Wild type (WT) mice and (**B**) $Cnr2^{-/-}$ mice treated with PBS (vehicle [Veh]), indomethacin (positive control), HU910 (a selective CB2 agonist and positive control), or OGP prior to xylene application. Ear swelling is presented as the difference between the xylene-treated and the untreated ear. Data are the mean ± SD, n=6. Color-coded * for $p<0.05$, Veh (turquoise) vs. indomethacin (gray), HU910 (blue), or OGP (green) treated mice at the same time point, two-way ANOVA. In either genotype, $p<0.001$ for the effect of xylene in the Veh group. (**C**) Photomicrographs of the mid-ear region from median representative mice.

The online version of this article includes the following source data for figure 7:

**Source data 1.** Ear edema in WT and Cnr2-/- mice.

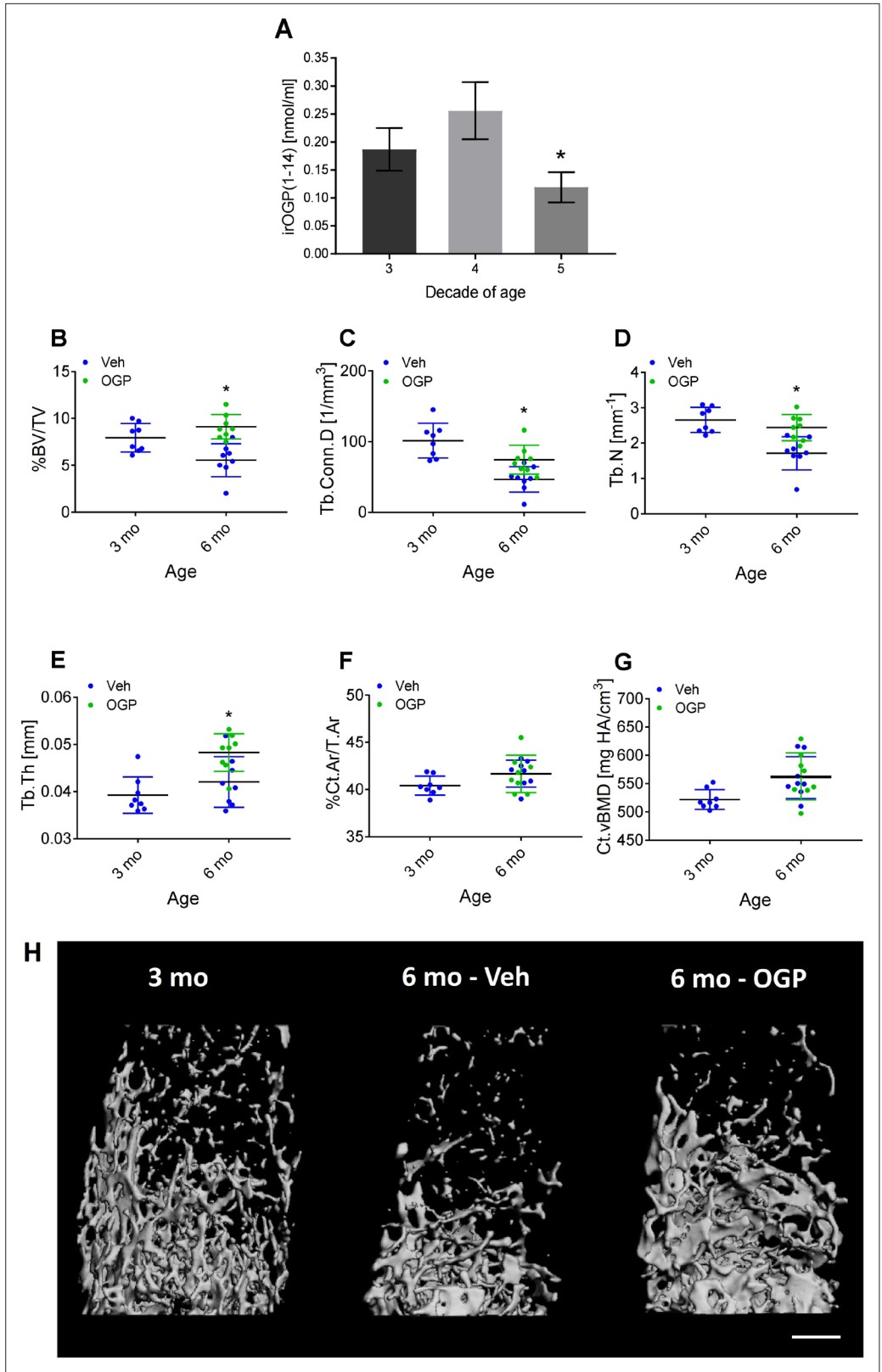

**Figure 8.** Osteogenic growth peptide (OGP[1-14]) levels in women and the effect of OGP on age-related bone loss in male mice. (**A**) Immunoreactive (ir) OGP(1-14) levels were measured in serum from human female subjects aged 18–49 years. irOGP(1-14) levels were significantly lower in the fifth decade compared with the third and fourth decades of life. Data are the mean ± SD. n=28 (third decade), n=6 (fourth decade), n=6 (fifth decade).

*Figure 8 continued on next page*

*Figure 8 continued*

*p<0.05 vs. fifth decade aged women, Kruskal-Willis oneway ANOVA and Mann Whitney-Wilcoxon rank sum tests. (**B–E**) Exogenous administration of OGP prevents age-related decline in (B) trabecular bone volume fraction (BV/TV), (C) trabecular connectivity density (Conn.D), (D) trabecular number (Tb.N) and (E) trabecular thickness (Tb.Th) in the distal femur of 3 month (3 mo) and 6 month old (6 mo) male mice. Data are the mean ± SD, n=8. *p<0.05 vs. vehicle (Veh)-treated mice, one-way ANOVA. (**H**) Representative μCT images of the distal femur from each group. Bar = 0.5 mm.

The online version of this article includes the following source data and figure supplement(s) for figure 8:

**Source data 1.** OGP serum levels in women.

**Source data 2.** MicroCT data for 3- and 6-mo mice treated with OGP.

**Figure supplement 1.** Effect of osteogenic growth peptide (OGP) on age-related cortical bone loss in male mice.

allosteric site, is hard to predict. Notably, the peptide is very flexible and its conformation is therefore rather elusive. In addition, in the case of the homology models, the overall structure of the receptor should also be considered with care. Hence, our results suggest that OGP can bind to the ECL region, although the exact binding mode may be slightly different than that obtained from our calculations. Our findings are supported by recent studies of Pandey et al. who also suggested a possible allosteric site for dihydro-gambogic acid and trans-β-caryophyllene within the CB2 receptor (*Pandey et al., 2020*). However, their study suggested a binding site that involved ECL1, ECL2, and the TM region, while our allosteric site for OGP involved only the ECL regions. These differences may arise from differences in the chemical character and activity of the studied ligands. More comprehensive studies are needed to determine whether OGP is a full agonist at CB2 or a PAM in conjunction with bound orthosteric agonists.

Many cannabinoid agonists bind to both CB1 and CB2, likely due to the moderate homology between CB1 and CB2. Following the recently reported crystal structure of CB2 (47), we estimate that these two receptors share a 30% identity (43% similarity) over the full sequence, 44% identity (62% similarity) without the N- and C-terminal regions, and 51% identity (69% similarity) if only the transmembrane regions are considered (*Supplementary file 3*; *Munk et al., 2019*). Assuming that the allosteric binding at CB2 involves the extracellular regions where the homology between the two receptors drops to 38% identity (or even 10% if we consider the N-terminal region) reduces the likelihood that OGP also binds to CB1. Indeed, using our cAMP assay in hCB1-transfected HEK293 cells, OGP failed to activate CB1 (*Figure 2—figure supplement 2*). However, we cannot rule out the possibility that under certain conditions OGP binds to CB1. More experiments are needed to elucidate this issue.

In vitro, all our assays showed a biphasic dose-response, where below the optimal dose, the effect increases with the dose, but after this optimal dose, the effect diminishes gradually to reach the no treatment levels (*Figure 4A and B*) and sometimes even inverses the effect (*Figure 4—figure supplement 1*). This biphasic effect has been repeatedly reported for OGP, other CB2 agonists, and agonists of other GPCRs (*Ofek et al., 2006*; *Miguel et al., 2005*; *Ofek et al., 2011*; *Bab and Chorev, 2002*; *Smoum et al., 2015*; *Schattauer et al., 2019*). The reversal of the effect at high doses likely results from desensitization of the receptor (*Kelly et al., 2008*). Interestingly, such a biphasic effect in vivo was also reported for another CB2 agonist (*Smoum et al., 2015*), although the most common observation is a plateau after the optimal dose is reached (*Bab et al., 1992*; *Schattauer et al., 2019*).

The presence of such a stable and selective CB2 agonist and PAM is of great significance since the known EC ligands (AEA and 2-AG) are prone to fast enzymatic breakdown, bind to both CB1 and CB2 and elicit different effects on each receptor. This study portrays OGP as a third endogenous cannabinoid with long-lasting effects and CB2 selectivity. The latter is particularly relevant in the CNS thus avoiding the psychoactive effect associated with CB1 EC agonists. Because CB2 has been primarily associated with the immune and skeletal systems in mice and men, it suggests that the main roles of OGP (and its potential therapeutic effects) are in these two systems.

Ideally, genetic ablation of OGP could be used to assess the physiological role of OGP. However, these loss-of-function approaches to study the physiological significance of OGP are problematic. First, molecular targeting of OGP, such as by gene ablation or RNA interference, would also target histone H4 expression, which is indispensable for life. Second, gene replacement studies (knock-in) for histones are unrealistic, given their multiple copies in mammalian genomes and that deletion of

all these genes is not compatible with life (*Smith et al., 2005*). Therefore, in our approach for testing the OGP bone protective effect, we employed an experiment involving long-term treatment of OGP, supposedly compensating for the age-related decline of the endogenous peptide in the circulation. Using this approach, we were able to prevent age-related bone loss in mice, thus confirming the skeletal protective role of OGP. These data strengthen the assumption that the reduced levels of OGP are at least partially responsible for the slow decline in bone mass with age, starting after the age of peak bone mass.

An important part of this study was to assess the link between OGP and age-related bone loss. A link between OGP levels and age in men has already been published (*Greenberg et al., 1995*), as well as in non-OVX female mice overexpressing OGP (*Smith et al., 2005*). We show here new data on premenopausal women as well as OVX female and naive male mice treated with exogenous OGP (*Figures 5 and 8*, *Figure 5—figure supplement 1* and *Figure 8—figure supplement 1*) during an age range associated with age-related bone loss in both humans and mice (*Kralick and Zemel, 2020* and *Figure 1*). Collectively, the published literature (*Greenberg et al., 1995*; *Smith et al., 2005*) and this study indicate a link between age and endogenous OGP levels in men and women, and demonstrate that increasing OGP levels increases bone mass in both male and female mice.

Our findings prompted us to envision additional experiments to assess their translational potential. Since the biosynthetic pathway of OGP is relatively well defined, this pathway could be modulated (e.g. stimulate H4 leaky ribosomal scanning, enhance OGP[1-14] proteolytic activation) to maintain optimal OGP levels. The development of a controlled-release device that would normalize the serum levels of OGP throughout aging is also a realistic approach in managing and minimizing age-related bone loss. The specificity of this peptide agonist for CB2, if confirmed, would imply the absence of undesirable psychotropic effects shared by many of the non-specific cannabinoid receptor ligands that, in addition to CB2, activate neuronal CB1 receptors. Moreover, the results of our study will potentially pave the way for OGP-based treatments for a vast array of skeletal and inflammatory conditions. On a broader perspective, OGP is secreted by osteoblasts (*Bab et al., 1999*) and this is one of the very few examples attributing a systemic endocrine role to osteoblasts. From an osteoimmunological standpoint, previous studies showed mechanisms related to the effects of inflammation and immune cells on bone cells (*Bar-Shavit, 2008*; *Blaschke et al., 2018*), some show interactions between osteoclasts and hematopoietic/immune cells (*Kollet et al., 2006*), or cytokine expression in osteoblasts (e.g. *Patil et al., 2004*). However, whether osteoblasts may suppress inflammatory processes remains an open question. OGP may be the first example of an anti-inflammatory hormone secreted by osteoblasts and this study prompts future research on the pathophysiological and clinical relevance of this putative immunomodulatory role of bone cells.

## Materials and methods
### Bone cell cultures

New-born mouse calvarial osteoblasts were prepared from 5-day-old mice by successive collagenase digestion (*Smoum et al., 2010*). Human osteoblasts were obtained from the cancellous bone of the head of the femur from patients undergoing total hip replacement (Helsinki ethics approval 0063–12-TLV) as reported previously (*Gartland et al., 2012*). For proliferation assays, cells pooled from five to six mice or from one patient were plated in 24-well format in triplicate, grown to subconfluence in α-MEM supplemented with 10% fetal calf serum (FCS) and then serum-starved for 2 hr. Cell counts were determined after an additional 48 hr incubation in α-MEM supplemented with 0.5% BSA and the tested compound, OGP or HU910, with or without SR144528 (CB2 selective antagonist), at the indicated concentrations. Osteoclastogenic cultures were established from bone marrow-derived monocytes of 10–11 week-old mice and grown for 4–5 days in medium containing M-CSF (CMG[14-12] supernatant *Faccio et al., 2003*) and RANKL ([R&D Systems] as reported previously [*Asagiri and Takayanagi, 2007*]). Cells from two mice per genotype were pooled together and plated in six wells in a 96-well plate format. For osteoclast cell counts the osteoclastogenic cultures were fixed in formaldehyde and then stained for tartrate-resistant acid phosphatase (TRAP, Sigma) (*Smoum et al., 2010*). All in vitro experiments were replicated independently at least three times. The BrdU assay for the determination of osteoblast proliferation was performed as previously reported (*Miguel et al., 2005*).

## MD computational details

The crystal structure of the CB2 receptor, recently reported, presents an inactive conformation (*Li et al., 2019*), whereas that of CB1 receptor presents an active conformation (*Hua et al., 2017*). The two receptors belong to the same subfamily (of lipid-cannabinoid receptors). The percent identity and similarity between the different segments of the CB1 and CB2 show that they share about 44% identity (62% similarity) without the N- and C-terminal regions (*Supplementary file 3*). A homology model of active CB2 was therefore constructed based on alignment to the structure of CB1. The crystal structure and the homology model of CB2 are, thus, referred to as the inactive and active CB2 models, respectively. The active model was constructed based on alignment to the structure of CB1 (PDB ID: 5XRA) (*Hua et al., 2017*) after elimination of the fused flavodoxin, using GPCRdb in Modeller 9.19 (*Pándy-Szekeres et al., 2018*; *Webb and Sali, 2016*). The inactive model involved replacing the fused T4-lysozyme in the CB2 crystal structure (PDB ID: 5ZTY) (*Li et al., 2019*) with ICL3, reverting mutations back into WT residues and filling up the gaps in the structure. Docking calculations utilized Glide (*Friesner et al., 2004*) and MD simulations involved NAMD (*Phillips et al., 2005*). OPLS3 force field was applied in all docking calculations (*Harder et al., 2016*). Protonation states of titratable residues were adjusted to pH = 7.4. Two plausible binding sites within the model of the CB2 receptor were identified using the sitemap module of Schrödinger (*Halgren, 2009*). Ligand and peptide docking options of Glide were used to dock CP55940 and OGP, respectively. A cubical grid of 30 Å centered at the centroid of D25 and F183 was generated covering the two possible binding sites. The lowest energy conformations of peptide-receptor complex, which are expected to represent complexes with the highest binding propensity, were used to further understand the binding. The lowest energy conformation of OGP bound to the inactive model of CB2 at the ECL region in the presence and absence of CP55940 were subjected to MD simulations in NAMD (*Phillips et al., 2005*). Initial structure for MD has been prepared by http://www.charmm-gui.org web portal (*Jo et al., 2017*) using charmm36 force field (*Best et al., 2012*). Homogeneous (1-palmitoyl-2-oleoyl-sn-glycero-3-ph osphocholine) POPC lipid bilayer was built around the protein in a rectangular box. The protein-lipid complex was solvated using the TIP3P water model in an orthorhombic water-box with water height of ~30 Å distance above and below the lipid bilayer. A restrained minimization was carried out for 10,000 steps using the conjugate gradient method. The minimized system was then equilibrated for 100 ps while keeping protein and lipid molecules restrained followed by additional 600 ps gradually releasing these restraints. The temperature was set to 310.15 K. Finally, the system was propagated for 100 ns using NP$\gamma$T thermodynamic ensemble.

## Binding assay (cAMP measurement)

CHO-K1 cells stably transfected with cDNA encoding human *CNR2* (CHO-hCB2) were purchased from PerkinElmer Life Sciences. They were maintained at 37°C in 5% (vol/vol) $CO_2$ in Dulbecco's modified Eagle medium nutrient mixture F-12 HAM, supplemented with 1 mM L-glutamine, 10% (vol/vol) FCS, 0.6% penicillin–streptomycin and 400 μg/ml G418. To study binding to CB1, we transfected HEK293 cells with human *CNR1* (HEK-hCB1) with an expressing vector purchased from PerkinElmer (formerly Horizon/Open Biosystems). Cell lines were purchased from ATCC and were routinely tested for mycoplasma (EZ-PCR Mycoplasma Detection Kit, Biological Industries). We confirmed the absence of CNR1 and CNR2 expression in the untransfected and mock-transfected CHO-K1 and HEK293 and validated expression in the transfected CHO-hCB2 and HEK-hCB1 cells using RTqPCR; primers for human *CNR1* F-CCTAGATGGCCTTGCAGATAC, R-ACACTGGATGTTCTCCTCATTC and for human *CNR2*, F-GATTGGCAGCGTGACTATGA, R-GAGAACATGCCCATAGGTGTAG. The mock-transfected CHO-K1 and HEK293 cells as well as the CHO-hCB2 and HEK-CB1 cells were treated with OGP, HU-910 (CB2 agonist), HU-210 (CB1 agonist), or Veh (control), before adding FSK at a concentration of 10 μM (*Horváth et al., 2012*). Samples were analyzed using a colorimetric cAMP ELISA kit (Cayman Chemicals). Additional cell transfections, culture, and adenylate cyclase assay were carried out as previously described (*Bayewitch et al., 1995*; *Vogel et al., 1993*).

## Effect of OGP on the [35S]GTP$\gamma$S binding

Membranes prepared from CHO-KI cells expressing CB2 or control empty vectors (50 μg per well) were incubated with tritiated CP55940 and a range of concentrations of OGP. The method for measuring agonist-stimulated [35S]GTP$\gamma$S binding to CB2 was adapted from *Cascio et al., 2010*. The

assays were carried out with GTPɣS binding buffer (50 mM Tris-HCl, 50 mM Tris-Base, 5 mM MgCl2, 1 mM EDTA, 100 mM NaCl, 1 mM dithiothreitol, and 0.1% BSA) in the presence of [35S]GTPɣS and GDP, in a final volume of 500 ml. Binding was initiated by the addition of [35S]GTPɣS to the wells. Nonspecific binding was measured in the presence of 30 mM GTPɣS. The reaction was terminated by a rapid vacuum filtration method using Tris-binding buffer as described previously (*Cascio et al., 2010*), and the radioactivity was quantified by liquid scintillation spectrometry. Agonists were stored at –20°C as 10 mM stock solutions dissolved in distilled water or DMSO.

## In vitro inflammation

BMDMs were obtained from C57BL/6 J WT and Cnr2$^{-/-}$ female mice as described previously (*Asagiri and Takayanagi, 2007*). Briefly, bone marrow was flushed out and cells were plated on tissue culture treated plate overnight and non-adherent cells were then plated in non-tissue culture dishes in the presence of 100 ng/ml M-CSF for 7 days with medium renewal every 48 hr. Trypsin was used to disconnect the cells from the plates. The cells were then washed with phosphate-buffered saline, resuspended in α-MEM supplemented with 10% FCS, and plated (1.2×10$^5$) in six-well flat-bottom plates. Various concentrations of OGP, HU-910 or Veh were added to the macrophages, followed by the addition of 1 µg/ml of LPS(*Escherichia coli*) for activation. The macrophages were then incubated in a humid atmosphere with 5% CO$_2$ overnight. The mRNA was extracted from the cells to be assayed by real-time quantitative PCR (RT-qPCR) for *Tnfa* and *Il1b* gene expression using the following primer sequences: F-TCTTCTCATTCCTGCTTGTGG and R-GGTCTGGGCCATAGAACTGA for *Tnfa*, and F-GAAATGCCACCTTTTGACAGTG and R-TGGATGCTCTCATCAGGACAG for *Il1b*. The normalizing gene, *Gapdh*, primer sequence is F-GTCACCCACACTGTGCCCATC and R-CCGTCAGGCAGCTCATAGCTC. RT-qPCR programming: 40 cycles, at 95°C for 20 s, at 60°C for 20 s, and at 72°C for 25 s (step One).

## Ear swelling

To induce ear inflammation and swelling, a volume of 20 µl xylene was pipetted to the inner and outer aspect of the right external ear (*Smoum et al., 2015*). Next, 20 µg/kg HU910, used as a positive control for CB2 activation (*Horváth et al., 2012*), 1 µg/kg OGP, or the Veh solvent ethanol-emulfor-saline (1:1:18) was injected subcutaneously 24 hr before applying xylene. Then, 2 mg/kg indomethacin, used as a general positive control, was given 30 min prior to applying xylene. PBS was applied to the other ear. Ear thickness was measured using a Mitutoyo micrometer (Mitutoyo Japan) immediately before as well as 30, 90, and 150 min after applying xylene or PBS. Ear swelling was expressed as the difference between the xylene- and PBS-treated ears (*Kou et al., 2005*). Mice were euthanized after the last measurement, ears were collected, formaldehyde fixed, and haematoxylin-eosin stained.

## EC extraction and CB2 expression in bone

Here, we analyzed female and male mice at 3 and 6 months of age, at that time tibia and femur bones were harvested, the femurs were analyzed with µCT as described below, and the tibia was subjected for EC and RNA extraction. The EC profile was determined in the tibia quantified by LC/MS/MS. LC/MS/MS protocol for tissue extraction of EC procedure was adapted from *Wang et al., 2003*. The mRNA was extracted from the femur to be assayed by RT-qPCR for *Cnr2* gene expression using the following primer sequences: F-ACGGTGGCTTGGAGTTCAAC and R-GCCGGGAGGACAGGATAAT.

## Rescue of OVX-induced bone loss

To test the skeletal activity of OGP, 8–9 weeks old WT, *Cnr2*$^{-/-}$ mice, on a C57Bl/6 J background, were subjected to bilateral OVX. The animals were not further treated for the next 6 weeks to allow for a substantial amount of bone loss and the establishment of a new bone remodeling balance (*Gavish et al., 1997*). The animals were then subcutaneously injected once a day, for additional 6 weeks, with PBS (negative controls), or 10 ng/day of synthetic OGP (test compound). The femora were then subjected to bone structural analyses by microcomputed tomography using a µCT system (µCT 50, Scanco Medical AG, Switzerland) as described below (*Benito et al., 2005*).

## OGP prevention of age-related bone loss

Here, we aimed to demonstrate the effect of OGP on the prevention of age-related bone loss. We could not detect the serum levels of OGP, owing to the lack of high-affinity antibodies against the short OGP, and this peptide could not be neutralized either by using neutralizing antibodies or by knocking out the histone H4 gene. We therefore opted for an experimental setting where we administered exogenous OGP to compensate for the decline in the endogenous levels of this hormone over time. To this end, 3-month-old male mice were treated with 1 µg/kg of OGP or PBS (control group), once daily 5 days a week for three more months. At the age of 6 months, mice were euthanized, and bones were harvested and analyzed with µCT.

## Structural skeletal µCT analysis

Trabecular and cortical bone was examined in the femoral distal metaphysis and mid-diaphysis, respectively, as reported previously (*Ofek et al., 2006*; *Bajayo et al., 2005*; *Yirmiya et al., 2006*). Briefly, femora were collected, fixed in phosphate buffered formalin for 48 hr, and further kept in 70% ethanol. They were examined by a Scanco µCT50 (Scanco Medical AG, Switzerland) system. Scans were performed at an isotropic 10 µm nominal resolution, with 90 kVp energy, 114 mA intensity, and 1100 ms integration time. The mineralized tissues were differentially segmented by a global thresholding procedure (140 and 224 permil for the trabecular and cortical bone, respectively). The bone parameters were determined using a direct 3D approach and data are presented in accordance with the official nomenclature (*Bouxsein et al., 2010*).

## Age-related serum levels of OGP(1-14) in female subjects

Healthy female volunteers were selected after completing a detailed questionnaire related to their medical history. The exclusion criteria included the use of medications known to affect bone and mineral metabolism (other than oral contraceptives), and conditions such as diseases of the parathyroid, thyroid, kidney, and liver. In addition, subjects who had experienced blood loss or bone fractures within the last 6 months were excluded, since these conditions may affect the serum level of OGP(1-14). Subjects were given specific instructions regarding sleep, diet, and water consumption at 6 hr prior to the blood extraction. Subjects included 28 women aged 18–29 years, 6 women aged 30–39 years and 6 women aged 40–49 years. Blood samples, 5–10 ml in volume, were taken between 11:00 am and 12:30 pm. Samples were incubated at room temperature for 30 min, centrifuged for serum separation and kept frozen until testing.

## OGP(1-14) determination via radioimmunoassay (RIA)

RIA to determine irOGP(1-14) was carried out using rabbit anti-OGP(1-14) antibody raised against the C-terminal region of OGP(1-14), as described before (*Gavish et al., 1997*). This antiserum detects both the full-length OGP(1-14) and the truncated OGP forms of the peptide. This assay is based on competitive binding to the antibodies of native and radioiodinated peptide. The reaction mixture consisted of 50 µl of the test sample and 100 µl of each of the following solutions: 1:40 dilution of non-immune rabbit serum, 1:500 dilution of rabbit anti-OGP(1-14) antiserum, and $4 \times 10^4$ cpm of $(3-^{125}I[Tyr^{10}])$ OGP(1-14). After overnight incubation at room temperature, the mixture was supplemented with 1 ml of a 1:50 diluted solution of goat anti-rabbit IgG (Sigma Chemical Co., St. Louis, MO, catalog no. R0881) and 2.5% polyethylene glycol (Sigma Chemical, catalogue no. P2263). This was followed by additional 2 hr shaking at 4°C and centrifugation at 4000× g for 15 min. The total radioactivity in the pellet was estimated in a gamma counter and irOGP(1-14) levels were calculated. The RIA intra- and inter-assay precision was approximately 5%.

### Statistics

For a sample size greater than or equal to six and with normal distribution, differences between time/treatment groups were analyzed by Student's t-test (for two groups only) or ANOVA (for three groups or more). When significant differences were indicated by ANOVA, the group means were compared using the Fisher-LSD test for pairwise comparisons. Multiple group comparisons with n<6 were analyzed using non-parametric ANOVA. A p-value less than 0.05 was considered statistically significant. For the human serum analysis, we used both the Kruskal-Willis oneway ANOVA and the Mann Whitney-Wilcoxon rank sum tests.

## Study approval

Animals - C57BL/6 J mice were used in all experiments. All procedures involving animals were carried out in accordance with the institutional guidelines and were approved by the Institutional Animal Care and Use Committee of Tel Aviv University (permit number M-14–092) and the Hebrew University of Jerusalem (permit number MD-12-13458-3). *CB2* knockout (*Cnr2$^{-/-}$*) were generated and shipped from the University of Bonn (Germany) and bred in the respective animal facilities at the Hebrew University and Tel Aviv University (SPF unit).

Human osteoblasts - The cells were obtained from patients undergoing total hip replacement (Helsinki ethics approval 0063–12-TLV).

Human serum - The protocol was designed in accordance the institutional guidelines and with the approval of the Institutional Research Committee for Human Studies of the Hebrew University-Hadassah Medical Center.

We declare that a written informed consent was received from all participants prior to inclusion in this study.

## Acknowledgements

The authors wish to thank Professor Raphael Mechoulam for his important advice and for providing the HU910 compound. Funding: This work was supported by Israel Science Foundation (ISF) Grants No. 1822/12, 1367/12 and 1086/17, and by an American Society of Bone and Mineral Research (ASBMR) GAP award to Y.G. This work was carried out in partial fulfillment of the requirements for a Ph.D. degree for BR-M from the Sackler Faculty of Medicine, Tel Aviv University, Tel Aviv, Israel.

## Additional information

### Funding

| Funder | Grant reference number | Author |
| --- | --- | --- |
| Israel Science Foundation | 1822/12 | Yankel Gabet |
| Israel Science Foundation | 1367/12 | Yankel Gabet |
| Israel Science Foundation | 1086/17 | Yankel Gabet |
| American Society for Bone and Mineral Research | Gap award | Yankel Gabet |

The funders had no role in study design, data collection and interpretation, or the decision to submit the work for publication.

### Author contributions

Bitya Raphael-Mizrahi, Conceptualization, Data curation, Formal analysis, Investigation, Methodology, Validation, Visualization, Writing - original draft, Writing - review and editing; Malka Attar-Namdar, Formal analysis, Investigation, Methodology, Project administration; Mukesh Chourasia, Formal analysis, Methodology, Software, Visualization; Maria G Cascio, Formal analysis, Methodology; Avital Shurki, Data curation, Formal analysis, Investigation, Methodology, Resources, Software, Supervision; Joseph Tam, Formal analysis, Methodology, Resources; Moshe Neuman, Formal analysis, Investigation, Methodology; Neta Rimmerman, Formal analysis, Investigation, Methodology, Validation; Zvi Vogel, Investigation, Resources, Supervision; Arie Shteyer, Resources, Supervision; Roger G Pertwee, Formal analysis, Methodology, Resources, Supervision; Andreas Zimmer, Investigation, Methodology, Supervision; Natalya M Kogan, Formal analysis, Investigation, Methodology, Supervision; Itai Bab, Conceptualization, Formal analysis, Funding acquisition, Investigation, Methodology, Project administration, Resources, Supervision, Validation, Writing - original draft; Yankel Gabet, Conceptualization, Data curation, Formal analysis, Funding acquisition, Investigation, Methodology, Project administration, Resources, Software, Supervision, Validation, Visualization, Writing - original draft, Writing - review and editing

**Author ORCIDs**
Bitya Raphael-Mizrahi  http://orcid.org/0000-0001-8629-1088
Joseph Tam  http://orcid.org/0000-0002-0948-0093
Yankel Gabet  http://orcid.org/0000-0002-7494-0631

**Ethics**
Human osteoblasts - The cells were obtained from patients undergoing total hip replacement (Helsinki ethics approval 0063-12-TLV). Human serum - The protocol was designed in accordance the institutional guidelines and with the approval of the Institutional Research Committee for Human Studies of the Hebrew University-Hadassah Medical Centre. We declare that a written informed consent was received from all participants prior to inclusion in this study.
Animals - C57BL/6J mice were used in all experiments. All procedures involving animals were carried out in accordance with the institutional guidelines and were approved by the Institutional Animal Care and Use Committee of Tel Aviv University (permit number M-14-092) and the Hebrew University of Jerusalem (permit number MD-12-13458-3). Cnr2 knockout (Cnr2-/-) were generated and shipped from the University of Bonn (Germany) and bred in the respective animal facilities at the Hebrew University and Tel Aviv University (SPF unit).

**Decision letter and Author response**
Decision letter https://doi.org/10.7554/eLife.65834.sa1
Author response https://doi.org/10.7554/eLife.65834.sa2

---

# Additional files

### Supplementary files
• Supplementary file 1. Literature search results summary, indicating the results of a Pubmed search (keywords ['peptide' or 'protein'] and ['Gi protein' or 'G(i)' or 'GPCR' or 'G*coupled receptor'] and [osteoblast* or osteoclast* OR osteocyte*]) where the activator interacts with a Gi-GPCR (26 agonists in total). The 10 agonists that are also endogenous proteins/peptides are highlighted in bold fonts. PTX, pertussis toxin.

• Supplementary file 2. Glide docking score (in kcal/mol) obtained for binding OGP in active and inactive models of CB2.

• Supplementary file 3. Percent identity and similarity between CB1 and CB2 calculated using the online resource by *Munk et al., 2019*.

• Transparent reporting form

### Data availability
All data generated or analyzed during this study are included in the manuscript and supporting files. Source data files have been provided for Figures 1 and 2 and 4–8.

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
