## [Editor Report]

By combining pharmacological and mouse genetic genetics strategies the authors show a clear interaction between the cannabinoid receptor CB2 and osteogenic growth peptide (OGP) in the control of bone remodeling and bone mass. They document that OGP attenuates bone loss by maintaining a skeletal CB2 tone, and it does so by allosterically binding to the CB2 receptor. These novel observations should allow further investigations on cannabinoid based strategies for skeletal diseases.

---

## [Decision Letter]

**Decision letter after peer review:**

Thank you for submitting your article "Osteogenic growth peptide is a potent anti-inflammatory and bone preserving hormone via cannabinoid receptor type 2" for consideration by *eLife*. Your article has been reviewed by 2 peer reviewers, and the evaluation has been overseen by a Reviewing Editor and Mone Zaidi as the Senior Editor. The reviewers have opted to remain anonymous.

Summary:

This is a well-written manuscript exploring the function and receptor for a peptide called Osteogenic Growth Peptide (OGP), which is derived from the histone H4 gene. The authors provide convincing evidence that the CB2 cannabinoid receptor serves as the cellular target for osteoblasts and osteoclasts in mediating the bone mass-promoting effects of this peptide.

Essential Revisions:

1. The manuscript needs to be better organized. The introduction needs to be revised to help readers understand the significance of the study and the role of CB2 on bone protection.

2. For Figure 1A, detailed descriptions of "in-depth literature search" should be included in the manuscript, such as the source of literatures or summary of 96 proteins to justify the importance of OGP.

3. The authors find the BV/TV of mouse trabecular bones at 6 months old is significantly lower than the one at 3 months old. The bone mass peak is achieved around 4-6 months of age in most mouse strains (Jilka 2013, Bernard 2002, Mouse Phenome Database, http://phenome.jax.org/). The decreased BV/TV may be the result of enlarged TV, not BV. Also, BV/TV of trabecular bones is the only parameter to evaluate bone phenotype throughout this study. Therefore, the existing data are not sufficient to support the conclusion of the regulation of CB2 signaling on bone loss and bone protection. More measurements, such as BV, trabecular bone number, trabecular thickness, bone mineral density of both trabecular bones and cortical bones, should be included to make reliable conclusions.

4. It is not sufficient to validate OGP binding to CB2 by molecular dynamics simulation alone. The binding should be at least validated by CO-IP assay etc.

5. It is not well accepted to use the number of cells to validate cell proliferation. Moreover, the authors have not indicated whether the "% control" refers to live cells or total cells. Trypan blue staining or other cell viability assays should be included. Other assays such as MTT, BrdU etc., would be better than cell counting to assess cell proliferation.

6. Low concentrations of OGP show the effects on osteoblast proliferation and anti-inflammatory processes. But high concentrations reverse these phenotypes. More explanations need to be provided.

7. The bone and anti-inflammatory phenotypes in CB2-/- mice should be validated in this study.

8. The authors demonstrate that anti-inflammatory activity is a hallmark of CB2 agonists. But the anti-inflammatory effects of CB2 signaling is not provided in the experiment. The comparison between CB2 -/- and WT cells should be included in the experiment. The conclusion would be more convincing if protein levels are measured.

9. For Figure 7, the authors measure the OGP (1-14) levels in women at 18-49. But they have not evaluated bone volume and density in those women. And also, they use 3-month old and 6-month old male mice to validate the skeletal effect. Therefore, age and sex groups of humans are not comparable to the groups of mice. A human at the age of 50 is approximately equivalent to a mouse at the age of 15 months (Dutta and Sengupta 2016). Therefore, the equivalent age and sex of mice and humans would be better to validate the beneficial effects of OGP on bones.

10. For Figure 1, OGP should have been measured in the aging mice as a control to show significant declines relative to other CB2 agonists.

11. For figure 7, where are the complete set of micro CT data parameters? Moreover, bone histomorphometric parameters with MAR, etc. should have been performed.

12. For figure 7, a great control given that the CB2 KO mice have no bone phenotype would have been to give OGP to them and demonstrate a failure of bone mass increase.*Reviewer #1 (Recommendations for the authors):*

Raphael-Mizrahi et al., investigate the role of osteogenic growth peptide (OGP) as cannabinoid receptor type 2 (CB2) agonist on bone and anti-inflammatory processes by clinical data, in-vitro, and in-vivo experiments. The authors show that OGP stimulates osteoblast proliferation at low concentrations and inhibits osteoclastogenesis via CB2. It also alleviates the acute inflammatory response by using the ear-swelling mouse model. This study suggests OGP is a potential candidate to treat inflammation and age-related bone loss. These data provide a possible alternative therapy for osteoporosis.

The conclusions of this study, to some extent, are supported by data, but some concerns need to be considered and clarified.

1. The authors demonstrated in the introduction that the selection of OGP as a CB2 agonist is through an in-depth literature search, but there is no further description of "search" in the manuscript.

2. The main findings of this study is that OGP activates CB2 signaling to rescue age-related bone loss. This conclusion is based on the premise that the activation of CB2 signaling protects bones. However, the bone phenotype of CB2-/- mice is not included in the study. Only BV/TV is not sufficient to indicate bone volume and quality. Thus, the major finding is not fully supported by the existing data.

3. The authors claim that OGP is CB2 agonist and binds to CB2, but only molecular dynamics simulation is weak to substantiate this conclusion.

4. The authors conclude OGP affects osteoblast proliferation. However, the method of cell counting is not sufficient to assess cell proliferation. Also, the method is not applied rigorously. It is not clear whether "% Control" refers to live cells only or total cells including dead cells. Therefore, it is difficult to assess this conclusion.

5. CB2 signaling affects anti-inflammatory processes. However, the conclusion is not reliable due to the lack of comparison between CB2 -/- and WT cells in the experiment. Moreover, the high standard deviation in three independent experiments makes it hard to assess the reproducibility of the conclusion.

6. It is not clear the underlying relationship between bone protection and anti-inflammatory functions of OGP.*Reviewer #2 (Recommendations for the authors):*

Overall this is a well-written manuscript of a research study involving in silico investigations, in vitro investigations, in vivo studies in mice, as well as ex vivo studies from human samples. The main premise sought to explore the bone effects of a peptide called Osteogenic Growth Peptide (OGP), which is derived from the histone H4 gene.

The authors provide convincing evidence that the CB2 cannabinoid receptor serves as the cellular target for osteoblasts and osteoclasts in mediating the bone mass-promoting effects of this peptide. Much of this experimentation takes advantage of CB2 knockout mice, which interesting by themselves have no phenotype. Here their initial experimentation could use strengthening as traditional evaluations such as CFU-OB assays or the use of CB2 KO mice for OGP injection experiments is currently lacking.

The manuscript goes on in its investigation to demonstrate that OGP has more "traditional" anti-inflammatory actions, such as the suppression of inflammatory cytokines or tissue inflammation. The data presented in that regard was sufficient.

This reviewer remains positive about this paper and overall finds the topic fascinating. With some revision to strengthen experimentation, the manuscript would reach an appropriate level warranting publication.

---

## [Author Response]

Essential Revisions:1. The manuscript needs to be better organized. The introduction needs to be revised to help readers understand the significance of the study and the role of CB2 on bone protection.

We added a paragraph describing the significance of this research.

2. For Figure 1A, detailed descriptions of "in-depth literature search" should be included in the manuscript, such as the source of literatures or summary of 96 proteins to justify the importance of OGP.

We modified the text in the introduction to make it clearer that the initial literature search generated 96 articles (not proteins). We made a new table (Supplementary File 1) where all the 26 Gi-GPCR agonists described in these 96 articles are listed (with references) and emphasized the 10 that are also endogenous peptides/proteins.

3. The authors find the BV/TV of mouse trabecular bones at 6 months old is significantly lower than the one at 3 months old. The bone mass peak is achieved around 4-6 months of age in most mouse strains (Jilka 2013, Bernard 2002, Mouse Phenome Database, http://phenome.jax.org/). The decreased BV/TV may be the result of enlarged TV, not BV. Also, BV/TV of trabecular bones is the only parameter to evaluate bone phenotype throughout this study. Therefore, the existing data are not sufficient to support the conclusion of the regulation of CB2 signaling on bone loss and bone protection. More measurements, such as BV, trabecular bone number, trabecular thickness, bone mineral density of both trabecular bones and cortical bones, should be included to make reliable conclusions.

The peak trabecular BV/TV in the distal femur of the C57Bl/6J mouse strain is achieved at around 3 months in males and even earlier in females, based on our own published data as well as in an additional independent study using the same analysis protocol (1, 2). The main decline is shown until 6 months of age. After that, bone mass continues to decline but at a slower pace (1, 2). To address the possible bias of changes in TV rather than BV, we looked at the data and added the following sentence in the Results section:

“In 3 months, the trabecular BV/TV decreased by 61% and 75% in males and females, respectively, due to significant 64% and 70% decrease in the bone volume.” (line 145)

As suggested, we also added a more comprehensive microCT analysis to the supplemental data (Figure 1 —figure supplement 1).

4. It is not sufficient to validate OGP binding to CB2 by molecular dynamics simulation alone. The binding should be at least validated by CO-IP assay etc.

Molecular dynamics simulation alone is indeed not sufficient to validate that OGP binds to CB2. In this study, our conclusion is mainly supported by our several cAMP inhibition assays (Figure 2 and Figure 2 —figure supplementary 2 from the original manuscript, now Figure 2 —figure supplementary 1 and more data included in the revised version). This assay is widely considered a binding assay, i.e. the inhibition of cAMP demonstrates agonist binding to a Gi-GPCR. To further strengthen our conclusion, we now add data from a competitive binding assay between OGP and tritiated CP55940 and an assay that shows the effect of OGP on the binding of ^[35S]^GTPγS that is CP55940-induced (Figure 3). In these assays we show that OGP and CP55940 do not compete on the same binding site and that OGP has a positive allosteric effect via its binding to the receptor. Results from these assays further indicate that OGP binds to CB2 and strengthens the evidence that OGP is a positive allosteric modulator as suggested from the molecular dynamics findings. It is worth mentioning that we could not perform a CO-IP assay since there are no available antibodies against the 5-amino acid OGP. The antibodies that were used to produce the human data were custom-made against the 14-amino acid OGP and required a large amount of serum that couldn’t be extracted from a mouse (as mentioned in line 265-269 in the main manuscript). Work is in progress to develop a good antibody that will detect OGP in small amounts of serum and can be used for immunofluorescence.

5. It is not well accepted to use the number of cells to validate cell proliferation. Moreover, the authors have not indicated whether the "% control" refers to live cells or total cells. Trypan blue staining or other cell viability assays should be included. Other assays such as MTT, BrdU etc., would be better than cell counting to assess cell proliferation.

Indeed, the % control are referred to live cells only since the dead cells are washed out and only adherent live cells are counted. The data were normalized so that the control number of live cells are 100% and all data is in ratio to the control. Additionally, we performed BRDU assays and added the results to the supplemental data (Figure 4 —figure supplement 1) with reference to this assay and results in the text.

6. Low concentrations of OGP show the effects on osteoblast proliferation and anti-inflammatory processes. But high concentrations reverse these phenotypes. More explanations need to be provided.

GPCR are known to have a biphasic effect when a vast range of agonist concentration are used (3). This biphasic effect was demonstrated many times before in response to high vs. low concentrations of ligands of the endocannabinoid system (4). More explanation is also added to the main manuscript in the discussion (paragraph starting with “in vitro, all our assays showed a biphasic dose-response…”).

7. The bone and anti-inflammatory phenotypes in CB2-/- mice should be validated in this study.

The increased inflammatory phenotype in *Cnr2^-/^*^-^ has been repeatedly demonstrated (reviewed in (5)). The bone phenotype of *Cnr2^-/-^* mice has been comprehensively studied in our and others’ published articles (6-8). Here we show that CB2 agonists strongly suppress inflammation in the ear edema in vivo model and in macrophages in vitro, via the CB2 receptor. Regarding the bone phenotype, here we show that OGP affects osteoblast proliferation and osteoclast differentiation via CB2. In this revised version, we now add data on an ovariectomy-induced osteoporosis experiment in WT and *Cnr2^-/-^* mice. The results show that OGP rescues OVX-induced bone loss via CB2 (see Results, Figure 5 and Figure 5 —figure supplement 5).

8. The authors demonstrate that anti-inflammatory activity is a hallmark of CB2 agonists. But the anti-inflammatory effects of CB2 signaling is not provided in the experiment. The comparison between CB2 -/- and WT cells should be included in the experiment. The conclusion would be more convincing if protein levels are measured.

This study does not demonstrate that anti-inflammatory activity is a hallmark of CB2 agonists (see Introduction). This has been demonstrated by many others and is now a well-established evidence. In line with this established fact, here we show that CB2 agonists (OGP and HU910) strongly suppress inflammation in the ear edema in vivo model. We also show that OGP has an anti-inflammatory effect (decreased expression of TNFα and IL1β) in LPS-treated macrophages in vitro, via the CB2 receptor.

9. For Figure 7, the authors measure the OGP (1-14) levels in women at 18-49. But they have not evaluated bone volume and density in those women. And also, they use 3-month old and 6-month old male mice to validate the skeletal effect. Therefore, age and sex groups of humans are not comparable to the groups of mice. A human at the age of 50 is approximately equivalent to a mouse at the age of 15 months (Dutta and Sengupta 2016). Therefore, the equivalent age and sex of mice and humans would be better to validate the beneficial effects of OGP on bones.

The main purpose of the mouse and human experiments is to assess the link between OGP and age-related bone loss. A link between OGP levels and age in men has already been published (9); as well as in female mice overexpressing OGP (10). We therefore show here new data on pre-menopausal women and male mice treated with exogenous OGP (new Figure 8). We also added in this revised manuscript, data on OVX females treated with OGP (new figure 5). Collectively, the published literature (9, 10) and this study (Figure 8) indicate a link between age and endogenous OGP levels during the period associated with slow age-related bone loss in both men and women (11), and demonstrate that increasing OGP levels increases bone mass (Figure 8) during a period associated with age-related bone loss in both male and female mice (Figure 1). In our mouse strain, using older animals is problematic as beyond 6 months in C57BL/6J-Rcc mice, there is almost no trabecular bone that would allow for a meaningful evaluation of further age-related bone loss.

We added a paragraph to address this question in the discussion of the revised manuscript (one before last paragraph in the discussion, “An important part…”).

10. For Figure 1, OGP should have been measured in the aging mice as a control to show significant declines relative to other CB2 agonists.

We agree that this would have been preferable. Unfortunately, what was done in humans was made possible with a large volume of serum that could not be obtained from mice. All our attempts to develop a new anti-serum against OGP did not reach a high enough sensitivity for mouse serum. See also answer to Comment 4.

11. For figure 7, where are the complete set of micro CT data parameters? Moreover, bone histomorphometric parameters with MAR, etc. should have been performed.

In the revised manuscript we added a more comprehensive morphometric analysis of the cortical and trabecular parameters (for both the age-related and the OVX experiments, Figures 5, 8, Figure 5 —figure supplement 1 and figure 8 —figure supplement 1). Re histomorphometry, we did not inject calcein as this has been repeatedly examined in past publications (10, 12). At this point, we therefore cannot provide the dynamic histomorphometry without repeating the entire experiment and sacrificing a large number of animals.

12. For figure 7, a great control given that the CB2 KO mice have no bone phenotype would have been to give OGP to them and demonstrate a failure of bone mass increase.

We added an ovariectomy (OVX) induced bone loss experiment to show in vivo that OGP has no bone effect on *Cnr2^-/^*^-^ mice. The manuscript provides ample evidence in vivo and in vitro to demonstrate that all the bone protective and anti-inflammatory effect of OGP are dependent on CB2, including lack of binding and effect in CHO cells, osteoblasts, osteoclasts, macrophages and in vivo using inflammation (ear edema) and OVX-induced bone loss models. We hope the reviewers will agree that given the additional data provided in the revised manuscript, there is little added value to show again *Cnr2^-/-^* mice treated with OGP at this point of the study.

Reviewer #1 (Recommendations for the authors):Raphael-Mizrahi et al., investigate the role of osteogenic growth peptide (OGP) as cannabinoid receptor type 2 (CB2) agonist on bone and anti-inflammatory processes by clinical data, in-vitro, and in-vivo experiments. The authors show that OGP stimulates osteoblast proliferation at low concentrations and inhibits osteoclastogenesis via CB2. It also alleviates the acute inflammatory response by using the ear-swelling mouse model. This study suggests OGP is a potential candidate to treat inflammation and age-related bone loss. These data provide a possible alternative therapy for osteoporosis.The conclusions of this study, to some extent, are supported by data, but some concerns need to be considered and clarified.It is not clear the underlying relationship between bone protection and anti-inflammatory functions of OGP.

That’s an important question for which part of the answer might be philosophical. Still, we added two paragraphs in the discussion (last one and 4^th^ before last) to address this aspect from a biological and clinical perspective:

“The presence of such a stable and selective CB2 agonist […] Because CB2 has been primarily associated with the immune and skeletal systems in mice and men, it suggests that the main roles of OGP (and its potential therapeutic effects) are in these 2 systems.”

“On a broader perspective, OGP is secreted by osteoblasts (13) and this is one of the very few examples attributing a systemic endocrine role to osteoblasts. From an osteoimmunological standpoint, previous studies showed mechanisms relating to the effects of inflammation and immune cells on bone cells (14, 15), some show interactions between osteoclasts and hematopoietic/immune cells (16), or cytokine expression in osteoblasts (e.g. (17)). However, whether osteoblasts may suppress inflammatory processes remains an open question. OGP may be the first example of an anti-inflammatory hormone secreted by osteoblasts and this study prompts future research on the pathophysiological and clinical relevance of this putative immunomodulatory role of bone cells.”

Reviewer #2 (Recommendations for the authors):Overall this is a well-written manuscript of a research study involving in silico investigations, in vitro investigations, in vivo studies in mice, as well as ex vivo studies from human samples. The main premise sought to explore the bone effects of a peptide called Osteogenic Growth Peptide (OGP), which is derived from the histone H4 gene.The authors provide convincing evidence that the CB2 cannabinoid receptor serves as the cellular target for osteoblasts and osteoclasts in mediating the bone mass-promoting effects of this peptide. Much of this experimentation takes advantage of CB2 knockout mice, which interesting by themselves have no phenotype. Here their initial experimentation could use strengthening as traditional evaluations such as CFU-OB assays or the use of CB2 KO mice for OGP injection experiments is currently lacking.

Please see answers to comments 14 and 16 to the editor’s summary

The manuscript goes on in its investigation to demonstrate that OGP has more "traditional" anti-inflammatory actions, such as the suppression of inflammatory cytokines or tissue inflammation. The data presented in that regard was sufficient.This reviewer remains positive about this paper and overall finds the topic fascinating. With some revision to strengthen experimentation, the manuscript would reach an appropriate level warranting publication.

The authors wish to thank the Reviewers and the Editor for their thorough review of our work and the constructive comments that we feel significantly improved the quality of our manuscript.

References

1. Bab I, Müller R, Hajbi-Yonissi C, and Gabet Y. *Micro-Tomographic Atlas of the Mouse Skeleton.* New York: Springer; 2007.

2. Glatt V, Canalis E, Stadmeyer L, and Bouxsein ML. Age-related changes in trabecular architecture differ in female and male C57BL/6J mice. *J Bone Miner Res.* 2007;22(8):1197-207.

3. Schattauer SS, Bedini A, Summers F, Reilly-Treat A, Andrews MM, Land BB, et al. Reactive oxygen species (ROS) generation is stimulated by kappa opioid receptor activation through phosphorylated c-Jun N-terminal kinase and inhibited by p38 mitogen-activated protein kinase (MAPK) activation. *J Biol Chem.* 2019;294(45):16884-96.

4. Smoum R, Baraghithy S, Chourasia M, Breuer A, Mussai N, Attar-Namdar M, et al. CB2 cannabinoid receptor agonist enantiomers HU-433 and HU-308: An inverse relationship between binding affinity and biological potency. *Proc Natl Acad Sci U S A.* 2015;112(28):8774-9.

5. Turcotte C, Blanchet MR, Laviolette M, and Flamand N. The CB2 receptor and its role as a regulator of inflammation. *Cell Mol Life Sci.* 2016;73(23):4449-70.

6. Idris AI, van 't Hof RJ, Greig IR, Ridge SA, Baker D, Ross RA, et al. Regulation of bone mass, bone loss and osteoclast activity by cannabinoid receptors. *Nat Med.* 2005;11(7):774-9.

7. Idris AI, Sophocleous A, Landao-Bassonga E, van't Hof RJ, and Ralston SH. Regulation of bone mass, osteoclast function, and ovariectomy-induced bone loss by the type 2 cannabinoid receptor. *Endocrinology.* 2008;149(11):5619-26.

8. Ofek O, Karsak M, Leclerc N, Fogel M, Frenkel B, Wright K, et al. Peripheral cannabinoid receptor, CB2, regulates bone mass. *Proc Natl Acad Sci U S A.* 2006;103(3):696-701.

9. Greenberg Z, Chorev M, Muhlrad A, Shteyer A, Namdar-Attar M, Casap N, et al. Structural and functional characterization of osteogenic growth peptide from human serum: identity with rat and mouse homologs. *J Clin Endocrinol Metab.* 1995;80(8):2330-5.

10. Smith E, Meyerrose TE, Kohler T, Namdar-Attar M, Bab N, Lahat O, et al. Leaky ribosomal scanning in mammalian genomes: significance of histone H4 alternative translation in vivo. *Nucleic Acids Res.* 2005;33(4):1298-308.

11. Kralick AE, and Zemel BS. Evolutionary Perspectives on the Developing Skeleton and Implications for Lifelong Health. *Front Endocrinol (Lausanne).* 2020;11:99.

12. Bab I, Gazit D, Chorev M, Muhlrad A, Shteyer A, Greenberg Z, et al. Histone H4-related osteogenic growth peptide (OGP): a novel circulating stimulator of osteoblastic activity. *EMBO J.* 1992;11(5):1867-73.

13. Bab I, Gavish H, Namdar-Attar M, Muhlrad A, Greenberg Z, Chen Y, et al. Isolation of mitogenically active C-terminal truncated pentapeptide of osteogenic growth peptide from human plasma and culture medium of murine osteoblastic cells. *J Pept Res.* 1999;54(5):408-14.

14. Bar-Shavit Z. Taking a toll on the bones: regulation of bone metabolism by innate immune regulators. *Autoimmunity.* 2008;41(3):195-203.

15. Blaschke M, Koepp R, Cortis J, Komrakova M, Schieker M, Hempel U, et al. IL-6, IL-1beta, and TNF-α only in combination influence the osteoporotic phenotype in Crohn's patients via bone formation and bone resorption. *Adv Clin Exp Med.* 2018;27(1):45-56.

16. Kollet O, Dar A, Shivtiel S, Kalinkovich A, Lapid K, Sztainberg Y, et al. Osteoclasts degrade endosteal components and promote mobilization of hematopoietic progenitor cells. *Nat Med.* 2006;12(6):657-64.

17. Patil C, Zhu X, Rossa C, Jr., Kim YJ, and Kirkwood KL. p38 MAPK regulates IL-1beta induced IL-6 expression through mRNA stability in osteoblasts. *Immunol Invest.* 2004;33(2):213-33.